# OSMR controls glioma stem cell respiration and confers resistance of glioblastoma to ionizing radiation

Ahmad Sharanek[1,7], Audrey Burban[1,7], Matthew Laaper[1,2], Emilie Heckel[3,4], Jean-Sebastien Joyal[3,4], Vahab D. Soleimani[1,5] & Arezu Jahani-Asl [1,2,6✉]

Glioblastoma contains a rare population of self-renewing brain tumor stem cells (BTSCs) which are endowed with properties to proliferate, spur the growth of new tumors, and at the same time, evade ionizing radiation (IR) and chemotherapy. However, the drivers of BTSC resistance to therapy remain unknown. The cytokine receptor for oncostatin M (OSMR) regulates BTSC proliferation and glioblastoma tumorigenesis. Here, we report our discovery of a mitochondrial OSMR that confers resistance to IR via regulation of oxidative phosphorylation, independent of its role in cell proliferation. Mechanistically, OSMR is targeted to the mitochondrial matrix via the presequence translocase-associated motor complex components, mtHSP70 and TIM44. OSMR interacts with NADH ubiquinone oxidoreductase 1/2 (NDUFS1/2) of complex I and promotes mitochondrial respiration. Deletion of OSMR impairs spare respiratory capacity, increases reactive oxygen species, and sensitizes BTSCs to IR-induced cell death. Importantly, suppression of OSMR improves glioblastoma response to IR and prolongs lifespan.

[1] Lady Davis Institute for Medical Research, Jewish General Hospital, 3755 Chemin de la Côte-Sainte-Catherine, Montréal, QC H3T 1E2, Canada. [2] Integrated program in Neuroscience, Montreal Neurological Institute, 3801 University Street, Montréal, QC H3A 2B4, Canada. [3] Departments of Pediatrics, Pharmacology and Ophthalmology, Université de Montréal, CHU Sainte-Justine, Montréal, QC H3T 1C5, Canada. [4] Department of Pharmacology and Therapeutics, McGill University, Montréal, QC H3G 1Y6, Canada. [5] Department of Human Genetics, McGill University, 3640 Rue University, Montréal, QC H3A OC7, Canada. [6] Gerald Bronfman Department of Oncology and Division of Experimental Medicine, McGill University, 5100 Maisonneuve Blvd West, Suite 720, H4A3T2, Montréal, QC, Canada. [7] These authors contributed equally: Ahmad Sharanek and Audrey Burban. ✉email: arezu.jahani@mcgill.ca

Glioblastoma is the most malignant primary brain tumor in adults. The present standards of care include maximal surgical removal of the tumors followed by treatment with temozolomide (TMZ) and ionizing radiation (IR)[1–3]. Despite these intense efforts, tumor recurrence and therapeutic resistance leave patients with a median survival rate of less than 18 months following diagnosis. Identification of brain tumor stem cells (BTSCs) in glioblastoma tumors has transformed our understanding of tumor resistance to therapy and disease pathogenesis[4,5]. BTSCs can undergo self-renewal to replenish themselves or give rise to all the cellular subpopulations within a tumor to recapitulate the functional heterogeneity of the tumor. BTSCs are highly proliferative but at the same time show resistance to DNA damaging chemotherapy and IR therapy, raising the possibility that during treatment of glioblastoma tumors, a BTSC may generate a cellular hierarchy that contributes to the acquisition of drug resistance[6–8].

Metabolic reprogramming is a hallmark of glioma and contributes to drug resistance[9,10]. A classic metabolic shift that provides the tumor cells with a survival advantage is their adaptation to aerobic glycolysis, characterized by high glucose uptake, low oxygen consumption and high lactate production[9,11,12]. However, BTSCs possess a unique metabolic phenotype, with a distinct upregulation of oxidative phosphorylation (OXPHOS) and a low glycolytic rate[13]. This metabolic profile of BTSCs resembles that of neurons, as opposed to the majority of cells in the bulk of the tumor that rely on aerobic glycolysis[13]. Identification of metabolic vulnerabilities and their targeting in BTSCs provides a promising approach to overcome glioblastoma resistance to therapy.

Oncostatin M receptor (OSMR) is a member of the interleukin-6 receptor family that carries out a diverse range of cellular functions, including regulation of homeostasis, cell growth, and differentiation[14,15]. OSMR is expressed in many tumor cell types, including sarcoma, melanoma, glioma, breast, and prostate carcinoma[16]. Oncostatin M (OSM), the ligand for OSMR, is also reported to regulate different hallmarks of cancer[17,18]. OSM is shown to increase tumor growth and metastasis of prostate and breast cancer[17,19], and may promote epithelial-mesenchymal transition[17]. The expression of OSMR is upregulated in mesenchymal and classical glioblastoma subtypes and upregulation of OSMR correlates significantly with poor patient prognosis[20,21]. Previous studies have established that OSMR is significantly upregulated in human BTSCs that harbor the oncogenic epidermal growth factor receptor variant III (EGFRvIII)[20]. OSMR forms a co-receptor complex with EGFR-vIII to amplify receptor tyrosine kinase signalling and glioblastoma tumorigenesis. Gene expression profiling using RNA-Seq analyses of OSMR and EGFRvIII in mouse astrocytes revelated two gene sets: OSMR/EGFRvIII common and OSMR unique candidate target genes that were not shared by EGFR-vIII[20], suggesting that OSMR may regulate glioblastoma tumorigenesis via additional mechanisms. Here, we report our discovery of a mitochondrial OSMR that functions to maintain mitochondrial respiration independently of EGFRvIII. Deletion of OSMR impairs OXPHOS, promotes generation of reactive oxygen species (ROS), and induces cell death. Importantly, deletion of OSMR is sufficient to sensitize the response of glioblastoma tumors to IR therapy and to prolong lifespan.

## Results

### Presence of a mitochondrial OSMR in human BTSCs. To gain mechanistic insights into OSMR signalling network, we aimed to characterize the full landscape of OSMR interactome by employing immunoprecipitation (IP) coupled with mass spectrometry (IP-LC-MS/MS). Since endogenous OSMR expression level is significantly elevated in tumor cells that harbor EGFRvIII mutation[20], we employed EGFRvIII-expressing mouse astrocytes in order to identify potential OSMR binding partners endogenously using a specific antibody to OSMR. IP-LC-MS/MS analysis revealed a large cohort of mitochondrial proteins that are known to regulate electron transport chain (ETC) as well as mitochondrial respiration (Supplementary Table 1), raising the question of whether OSMR is targeted to the mitochondria. To address this question, we assessed possible presence of OSMR at the mitochondria by first performing cell fractionation on four different patient-derived BTSC lines. Across all the BTSCs tested, we observed the presence of OSMR in the mitochondrial fractions, with no cross contamination from the nuclear or the cytoplasmic fractions (Fig. 1a–d). We also assessed dose dependency in the localization of OSMR to the mitochondria via immunoblotting of different concentrations of mitochondrial fractions relative to the cytoplasmic fraction (Supplementary Fig. 1a, b). To examine that the presence of OSMR in the mitochondria was not due to the contamination of mitochondrial fractions with the plasma membrane or the mitochondria-associated endoplasmic reticulum (ER) membrane, all blots were re-probed with the plasma membrane protein, Na+/K+ ATPase, and the ER integral membrane protein, calnexin (Fig. 1a–d). Together, our results confirmed the presence of a mitochondrial OSMR. In another independent set of studies, we performed confocal imaging on two patient-derived EGFRvIII-expressing human BTSCs (#73 and #147) co-stained with antibodies to OSMR and the mitochondrial marker, ATP synthase inhibitory factor subunit 1 (ATPIF1). We observed that OSMR was found in puncti with ATPIF1 (Fig. 1e, f). Next, we employed proximity ligation assay (PLA) to assess protein-protein interaction in situ. Primary antibodies to OSMR and ATPIF1 were used to perform PLA in BTSC73 and BTSC147 (Fig. 1g, h), and the cells were further subjected to staining with the MitoTracker (Fig. 1i). Strikingly, we detected significant PLA signal in the mitochondria of BTSCs compared to controls in which the primary antibodies were omitted. In follow up studies, we designed a fluorescence recovery after photobleaching (FRAP) assay using a GFP-tagged human OSMR to examine the recruitment of the OSMR to the mitochondria. We generated BTSC73 expressing the fusion protein GFP-OSMR via lentiviral transduction. Cells were stained with MitoTracker and subjected to photobleaching of the GFP signal in select areas using a Zeiss LSM 800 confocal microscope. Time-lapse imaging revealed the recovery of the GFP signal, indicating the recruitment of the GFP-OSMR to the mitochondria as traced by overlapping of the signal with the MitoTracker (Fig. 1j). Together, using subcellular fractionation, high resolution confocal imaging, PLA and FRAP assays, we have established that OSMR is translocated to the mitochondria in human BTSCs.

### OSMR is translocated into the mitochondrial matrix. Our observation that OSMR is localized to the mitochondria led to the question of which mitochondrial compartment is enriched for OSMR. We treated mitochondrial fractions obtained from BTSCs with proteinase K (PK) or a combination of PK and Triton X-100. PK digests all the mitochondrial outer membrane proteins including TOM20 and BCL2, but not proteins that reside inside the mitochondria, such as prohibitin[22]. We found that, similar to prohibitin, OSMR was detected in PK-treated fractions (Fig. 2a, b), suggesting that OSMR is actively imported into the mitochondria, and it is either localized in the inner membrane, the intermembrane space, or the mitochondrial matrix. The IP-MS data revealed multiple mitochondrial proteins

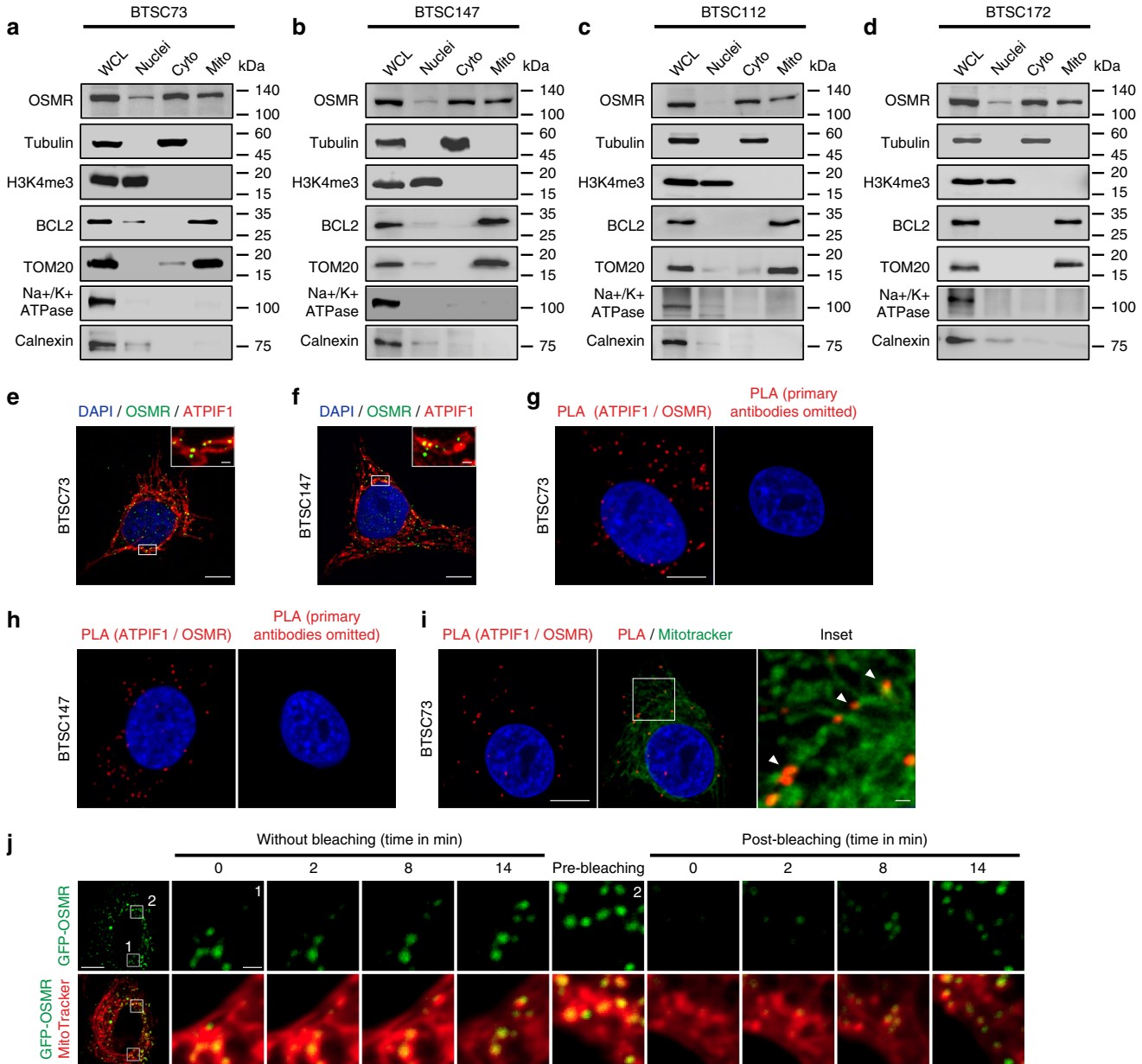

**Fig. 1 Presence of mitochondrial OSMR in human BTSCs. a–d** Four different patient-derived BTSC lines were subjected to subcellular fractionation, and the lysates for each fraction were analyzed by immunoblotting using antibodies to OSMR. α-Tubulin, H3K4me3, BCL2/TOM20, Na+/K+ ATPase, and calnexin. WCL: Whole-cell lysates; Cyto: cytoplasmic; Mito: mitochondrial. The Western blots represent a minimum of three replicates from different passage numbers for each BTSC. **e**, **f** BTSC73 and BTSC147 were subjected to immunostaining using antibodies to OSMR (green) and the mitochondrial matrix protein ATP synthase inhibitor F1 (ATPIF1, red). Nuclei were stained with DAPI. White rectangles mark the inset to demonstrate the co-localization of OSMR with ATPIF1. **g**, **h** PLA of OSMR and ATPIF1 were performed in BTSC73 (**g**) and BTSC147 (**h**). Primary antibodies were omitted as controls. **i** Double labeling of the PLA signal (red) and the MitoTracker (green) in BTSC73 is shown. **j** A FRAP assay was performed on BTSC73 transduced with GFP-OSMR and stained with MitoTracker (red). Different regions of interest (ROIs) containing GFP-OSMR in the mitochondria were defined. ROI1 indicates a non-bleached area and ROI2, a photobleached area. The fluorescence recovery was monitored over time following photobleaching. Images were obtained on a laser scanning confocal microscope (ZEISS LSM 800). Scale bar = 10 μm; Inset scale bar = 1 μm. Representative images of three independent experiments are shown.

including members of the presequence translocase-associated motor (PAM) import machinery, TIM44 and mtHSP70. Members of PAM complex are involved in the transport of mitochondrial matrix proteins into the mitochondrial matrix[23]. We performed co-immunoprecipitation (co-IP) followed by Western blot (WB) analysis, on either whole cell lysates (WCL) or purified mitochondrial fractions, and found that OSMR physically interacted with both mtHSP70 and TIM44 endogenously in multiple patient-derived BTSCs (Fig. 2c–f). Next, we performed

PLA analysis using antibodies to OSMR and mtHSP70. Significant PLA signal was detected in BTSC73 and 147 (Fig. 2g, h), and the PLA interaction signal was found in puncti with the mitochondria as revealed by co-staining of the BTSC73 with MitoTracker (Fig. 2i). In parallel experiments, we found that knockdown (KD) of either *mtHSP70* or *TIM44* significantly attenuated OSMR protein expression levels in the mitochondrial fractions (Fig. 2j, k). Our results suggest that OSMR is targeted to the mitochondrial matrix via PAM import machinery.

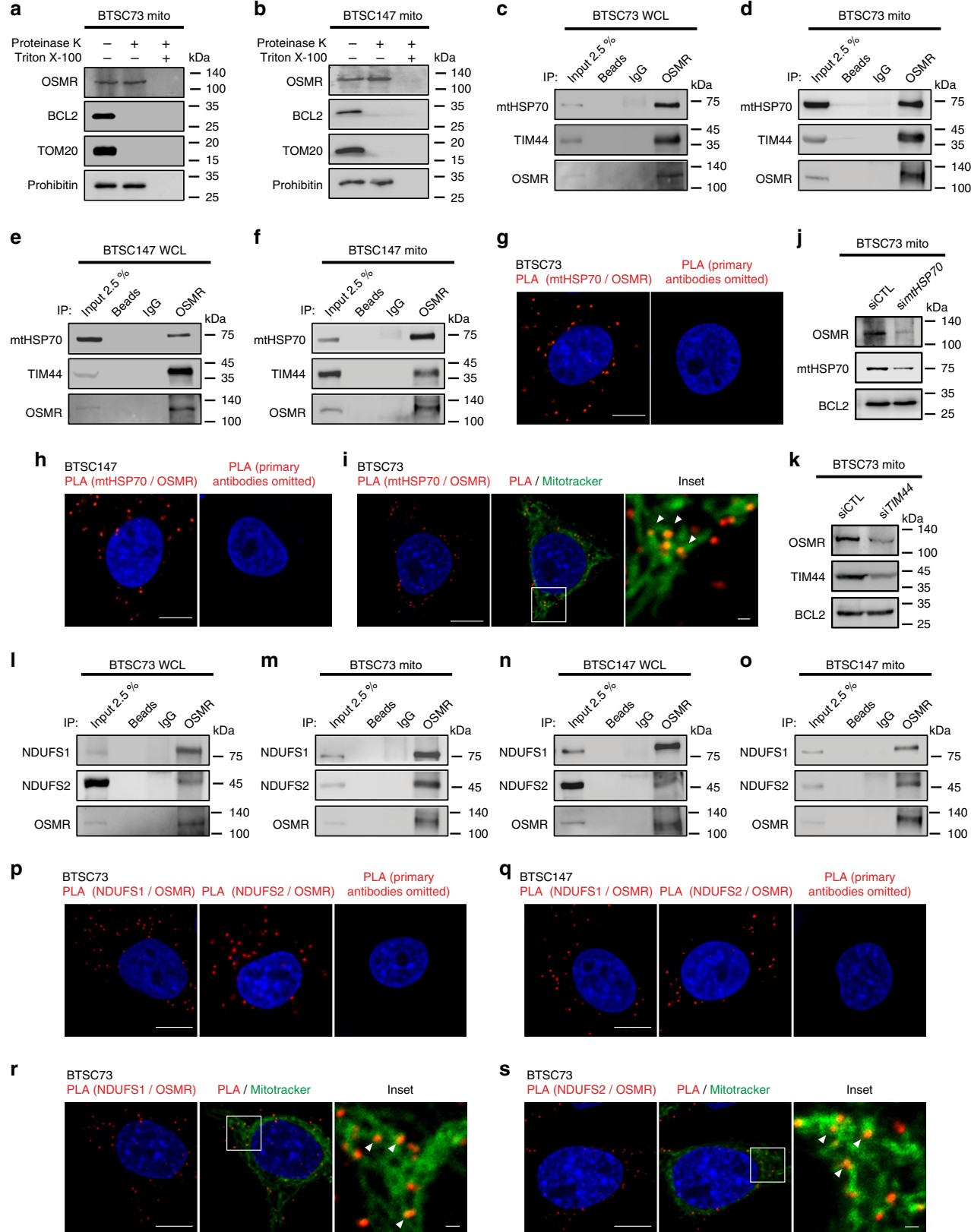

**OSMR interacts with the components of ETC.** In analysis of IP-LC-MS/MS data for potential OSMR binding partners in the mitochondrial matrix or the inner membrane, we identified NADH ubiquinone oxidoreductase 1/2 (NDUFS1/2) and NAD(P) transhydrogenase of mitochondrial complex I. NADH:ubiquinone oxidoreductase (complex I) is the largest complex of the mitochondrial ETC and contributes to ~40% of the proton motive force that is required for the mitochondrial ATP synthesis[24]. We asked if OSMR directly interacts with NDUFS1 and NDUFS2 in the mitochondria. Via co-IP of either WCL or purified mitochondrial fractions, we found that OSMR interacted endogenously with both NDUFS1 and NDUFS2 in multiple

**Fig. 2 OSMR interacts with different components of ETC in human BTSCs. a, b** Mitochondrial fractions from BTSC73 (**a**) and BTSC147 (**b**) were treated with 0.5 mg/mL proteinase K or proteinase K and 1% Triton X-100. Lysates were analyzed by immunoblotting using indicated antibodies. **c–f** WCL and mitochondrial fractions from BTSC73 (**c, d**) and BTSC147 (**e, f**) were subjected to immunoprecipitation using antibodies to OSMR or mouse IgG control, followed by immunoblotting with mtHSP70 and TIM44 antibodies. **g, h** PLA of OSMR and mtHSP70 were performed in BTSC73 (**g**) and BTSC147 (**h**). Primary antibodies were omitted for the controls. **i** Double labeling of the PLA signal (red) from the OSMR/mtHSP70 interaction and MitoTracker (green) is shown. **j** OSMR protein expression level was assessed in the mitochondrial fractions obtained from BTSC73 electroporated with siRNA control (siCTL) or siRNA against *mtHSP70* (si*mtHSP70*). BLC2 was used as a loading control. **k** OSMR protein expression level was assessed in the mitochondrial fractions obtained from BTSC73 electroporated with siCTL or siRNA against *TIM44* (si*TIM44*). BCL2 was used as a loading control. **l–o** WCL or mitochondrial fractions from BTSC73 (**l, m**) and BTSC147 (**n, o**) were subjected to immunoprecipitation using an antibody to OSMR or mouse IgG control followed by immunoblotting with NDUFS1 and NDUFS2 antibodies. **p, q** PLA analyses of OSMR/NDUFS1 and OSMR/NDUFS2 were carried out in BTSC73 (**p**) and BTSC147 (**q**). **r, s** Double labeling of the PLA signal (red) and the MitoTracker (green) is shown. Images were obtained with a 63X objectives on a laser scanning confocal microscope (ZEISS LSM 800). Scale bar = 10 μm. Inset scale bar = 1 μm. Representative images of three independent experiments are shown. The Western blots represent a minimum of three replicates from different passage numbers for each BTSC.

patient-derived BTSCs (Fig. 2l–o). In addition, the interactions of OSMR with either NDUFS1 or NDUFS2 were validated by PLA in different patient-derived BTSCs (Fig. 2p, q). Furthermore, we confirmed that the mitochondria is the site of interaction of OSMR/NDUFS1 and OSMR/NDUFS2 as revealed by double labeling of the PLA signal with the MitoTracker (Fig. 2r, s). Importantly, similar to our results in patient-derived BTSCs, we found that OSMR interacted with NDUFS1 and NDUFS2 as well as PAM complex components, mtHSP70 and TIM44, in different cell lines including the EGFRvIII-expressing mouse astrocytes (Supplementary Fig. 1c).

**OSMR controls ETC complex activities and ROS production.** The presence of OSMR in the mitochondria and its direct interaction with complex I core subunits led us next to investigate the functional consequences of these interactions. In analysis of public expression datasets[20], we found significant downregulation of mRNA of mitochondrial and metabolic genes including *Atp5b*, *Sirt3*, *Vdac3*, and *Atp6voc* in *OSMR* KD astrocytes. We induced *OSMR* KD in multiple patient-derived human BTSCs using CRISPR, CRISPRi, siRNA or shRNA approaches (Supplementary Fig. 2a–g). RT-qPCR analysis of the *OSMR* KD and control BTSCs revealed a significant reduction in mRNA expression levels of mitochondrial/metabolic genes including *ATP5B, SIRT3, VDAC3*, and *ATP6VOC* (Supplementary Fig. 2h–l), raising the question of whether OSMR regulates mitochondrial respiration and metabolism. To address this question, we first examined the specific enzymatic activities of complex I, II, III, IV, and ATP synthase in *OSMR* CRISPR and control BTSCs. The NADH: ubiquinone oxidoreductase (complex I) oxidizes NADH to facilitate the entry of electrons into the ETC and generates the proton gradient and ROS. We found a significant 28% decrease in the activity of complex I in *OSMR* CRISPR BTSCs compared to control (Fig. 3a). In assessing the succinate:ubiquinone oxidoreductase (complex II) activity, we also found a significant 30% decrease in *OSMR* CRISPR compared to control BTSCs (Fig. 3b). The flow of electrons generated by NADH and succinate oxidation via complex I and II is transferred to the complex III and IV. We assessed the activity of both the coenzyme Q-cytochrome c reductase (complex III) and the cytochrome c oxidase (complex IV), and noted a significant decrease of 35 and 23% in *OSMR* CRISPR BTSCs, respectively (Fig. 3c, d). In examining the ATP synthase activity in *OSMR* CRISPR and control BTSCs, however, we did not detect any significant changes (Fig. 3e). Our results suggest that OSMR impacts the activity of complex I to IV without affecting the ATP synthase.

Given that OSMR directly interacts with different components of complex I, and deregulation of complex I is a major source of mitochondrial ROS production[25], we next sought to examine whether OSMR regulates ROS levels in BTSCs. *OSMR* CRISPR

and control BTSCs were subjected to ROS analysis by flow cytometry using H2DCFDA assay. We found a significant increase in overall ROS levels in *OSMR* CRISPR BTSCs compared to control (Fig. 3f). In addition, we performed the MitoSOX-based flow cytometry assay to specifically assess the mitochondrial superoxide. Similar to the results obtained using H2DCFDA assay, we observed a significant increase in mitochondrial ROS levels in *OSMR* CRISPR BTSCs (Fig. 3g). Having established a role for OSMR in the regulation of ROS levels in vitro, we next examined whether OSMR regulates ROS levels in patient-derived tumors. *OSMR* CRISPR and CTL BTSC73 were injected into the flank of immunodeficient SCID mice subcutaneously and allowed to form tumors. The mice receiving the control BTSCs formed malignant tumors three weeks following injection while the mice receiving *OSMR* CRISPR BTSCs, had significantly smaller tumors. The tumor sections from each group were collected and analyzed using OxyIHC oxidative stress detection kit. We found a strong signal for OxyIHC staining in *OSMR* CRISPR small tumors compared to control malignant tumors (Fig. 3h).

**OSMR regulates mitochondrial oxygen consumption rate (OCR).** To determine the impact of OSMR on the overall mitochondrial respiration, we examined OCR as a measure of ETC activity using a Seahorse XFe96 Bioenergetic Flux Analyzer. We generated OSMR-overexpressing BTSC73 and subjected the cells to a mito stress test. We found a significant increase in cellular respiration in OSMR over-expressing cells compared to RFP-expressing control (Fig. 3i). Conversely, a significant reduction in cellular respiration was observed in BTSC73 *OSMR* CRISPRi compared to non-targeting gRNA control (Fig. 3j). In other experiments, *OSMR* KD BTSCs, generated by the delivery of two different *OSMR* shRNA, were subjected to mito stress tests. Consistent with the results obtained with *OSMR* CRISPRi model, we found a significant reduction in cellular respiration in BTSCs transduced with *OSMR* shRNA compared to scramble control (Fig. 3k and Supplementary Fig. 3a). Furthermore, KD of *OSMR* resulted in a significant increase in extracellular lactate levels (Supplementary Fig. 3b–d), suggesting that BTSCs upregulate glycolysis in response to *OSMR* KD. Together, our results established that OSMR physically and functionally interacts with the components of ETC and loss of OSMR impairs OXPHOS and generates ROS.

**The role of OSM/OSMR in respiration is independent of EGFRvIII.** OSMR orchestrates a feed forward signalling mechanism with the oncogenic protein, EGFRvIII and the transcription factor, signal transducer and activator of transcription 3 (STAT3) to drive cell proliferation[20]. Thus, we asked if OSM/OSMR-mediated regulation of cellular respiration is

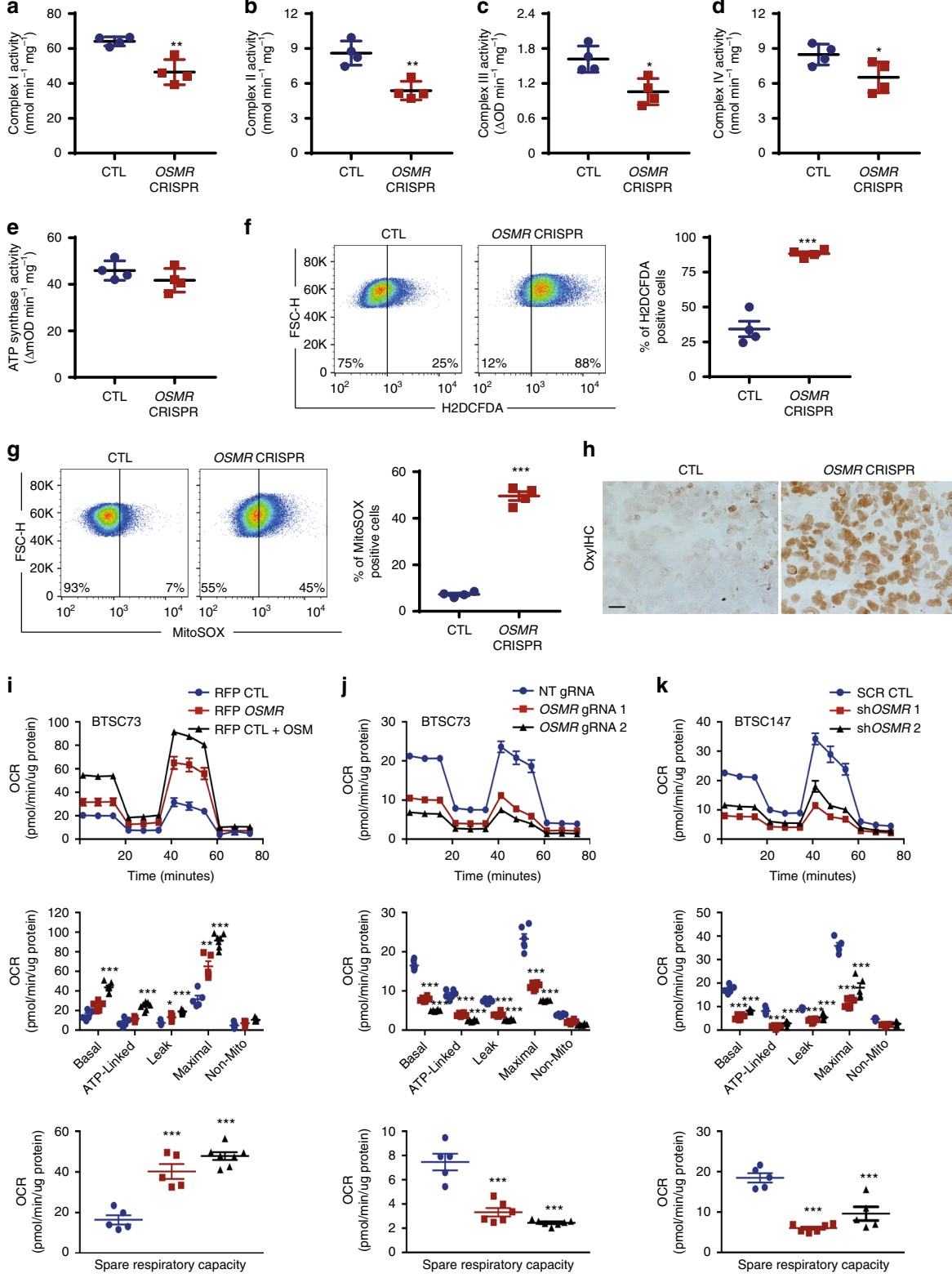

dependent on EGFRvIII and its role in proliferation. Addition of the ligand OSM to human BTSCs that naturally harbor EGFRvIII mutation from two patients (#73 and #147) induced a significant increase in respiration (Fig. 4a, b). Strikingly, BTSC12 and BTSC145 that do not harbor EGFRvIII mutation also displayed a robust increase in OCR in response to OSM (Fig. 4c, d), suggesting that EGFRvIII is not required for OSM-mediated upregulation of respiration. Given that OSM

signalling is also reported to induce the Janus Kinase (JAK)/ STAT, mitogen activated protein kinase (MAPK) and PI3K/ AKT signalling pathways[20,26–28], we next asked if any of these key oncogenic signalling pathways impacts OSM/OSMR induced mitochondrial respiration. To address this question, we first examined whether STAT3, p44/42 MAPK and PI3K/AKT signalling pathways are activated in response to OSM in patient-derived BTSCs, and whether pharmacological inhibitors

**Fig. 3 OSMR regulates mitochondrial OXPHOS and ROS generation. a–e** Enzymatic activities of mitochondrial ETC were analyzed in *OSMR* CRISPR or control BTSC73. Complex I, \*\**p* = 0.0037 (**a**); Complex II, \*\**p* = 0.0027 (**b**); Complex III, \**p* = 0.0126 (**c**); Complex IV, \**p* = 0.0488 (**d**); ATP synthase, *p* = 0.2506 (**e**); Unpaired two-tailed *t*-test, *n* = 4. **f** ROS generation was measured by flow cytometry using H2DCFDA in *OSMR* CRISPR and control BTSC73. \*\*\**p* < 0.0001; Unpaired two-tailed *t*-test, *n* = 4. **g** Mitochondrial superoxide abundance was assessed by flow cytometry using MitoSOX in *OSMR* CRISPR and control BTSC73. \*\*\**p* < 0.0001; Unpaired two-tailed *t*-test, *n* = 4. **h** Tumor sections from *OSMR* CRISPR and control BTSC73 were subjected to staining using OxyIHC oxidative stress detection kit. Representative images of 4 different tumor sections are shown. Scale bar = 20 μm. **i** RFP *OSMR* or RFP control BTSC73 (in the absence and presence of OSM) were subjected to bioenergetic analysis using a Seahorse XFe96 Bioenergetic Flux Analyzer. Oxygen consumption rates (OCR) are plotted (top panel). Data is plotted to demonstrate the differences between basal, ATP-linked, proton leak, maximal, and non-mitochondrial (mito) respiration (middle panel). Spare respiratory capacity (SRC), which is maximal minus basal respiration, is plotted (bottom panel). \*\*\**p* < 0.0001 for each pairwise comparison except: \**p*~Leak~ (RFP CTL vs. RFP *OSMR*) = 0.0225, \*\**p*~Maximal~ (RFP CTL vs. RFP *OSMR*) = 0.0021, \*\*\**p*~SRC~ (RFP CTL vs. RFP *OSMR*) = 0.0004; One-way ANOVA followed by Dunnett's test, *n* ≥ 5. **j** BTSC73 transfected with non-targeting (NT) gRNA control and two different gRNAs against *OSMR* were subjected to bioenergetic analysis as described in **i**. \*\*\**p* < 0.0001; One-way ANOVA followed by Dunnett's test, *n* ≥ 5. **k** BTSC147 transduced with two different *OSMR* shRNA (sh*OSMR* 1 and sh*OSMR* 2) and scramble shRNA control (SCR CTL) were subjected to bioenergetic analysis as described in **i**. \*\*\**p* < 0.0001 for each pairwise comparison except: \*\*\**p*~SRC~ (CTL vs. sh*OSMR* 2) = 0.0003; One-way ANOVA followed by Dunnett's test, *n* ≥ 5. Data are presented as the mean ± SEM. *n* represents an independent biological sample.

of these pathways including PD0325901 (MAPK inhibitor), LY294002 (PI3K/AKT inhibitor), and WP1066 (STAT3 inhibitor) impair the phosphorylation events in response to OSM. We found that OSM induced the phosphorylation of MAPK (Thr202/Tyr204), AKT (Ser473), and STAT3 (Tyr705), and the phosphorylation levels of these proteins were profoundly reduced in response to the pharmacological inhibitors (Supplementary Fig. 4a–c). Next, we examined if any of these inhibitors had an impact in reducing mitochondrial respiration in response to OSM. We measured OCR using a Seahorse XFe96 Bioenergetic Flux Analyzer in OSM-treated BTSC73 in the absence and the presence of the inhibitors. While PD0325901 induced an 11.3% decrease in maximal respiration in response to OSM, LY294002 attenuated respiration by 16.3% (Fig. 4e, f). In response to WP1066, however, we observed a 17% decrease in maximal respiration, 20% decrease in basal respiration, and 25% decrease in ATP-linked respiration, without a significant decrease in SRC (Fig. 4g). These differences can be attributed to the direct impact of STAT3 on mitochondrial respiration[29], or potentially due to the attenuated levels of OSMR expression in response to STAT3 inhibition[20]. In summary, our results suggest that OSM significantly induces mitochondrial respiration even after taking into consideration the impact of MAPK, PI3K/AKT, and STAT3 signalling pathways.

**The role of OSM/OSMR in respiration is independent of proliferation.** In view of our results that EGFRvIII, MAPK, and PI3K/AKT signalling pathways are not required for OSM regulation of mitochondrial respiration, we sought to determine whether OSMR regulation of cellular respiration is independent of its role in proliferation. First, we asked if translocation of OSMR to the mitochondria is a conserved mechanism in other tissues and model systems. Fractionation of wild type mouse tissues, including brain and liver, revealed that OSMR is targeted to the mitochondria in normal tissues from 8-week-old wild type mice (Supplementary Fig. 5a–c). Importantly, similar to BTSCs, we established that OSMR interacted with the components of complex I and PAM complex in post-mitotic primary cerebellar granule neurons (CGN) in which OSMR was ectopically expressed (Supplementary Fig. 5d). We determined the metabolic activity of post-mitotic CGN in response to OSM treatment using Seahorse bioenergetic analysis. Interestingly, similar to BTSCs, OSM induced a significant increase in respiration in post-mitotic neurons (Fig. 4h). Our data confirm that OSM/OSMR regulation of mitochondrial respiration is independent of its role in cell proliferation, and this mechanism is operational in post-mitotic

neurons upon forced induction of OSM/OSMR signalling pathways.

**OSMR controls BTSC self-renewal and confers resistance to IR.** Elevated levels of ROS is shown to sensitize cancer stem cells and tumor response to IR[30–32]. Our results demonstrated that inhibition of OSM/OSMR signalling pathway profoundly increased ROS levels, in vitro and in vivo. This led to the question of whether OSMR confers resistance to IR in human BTSCs. We performed limiting dilution assay (LDA) and extreme limiting dilution assay (ELDA)[33] to examine the response of BTSC73 and BTSC147 to IR in the absence and presence of OSM. OSM significantly protected against IR-induced cell death with a robust increase in the number of spheres surviving following IR treatment (Fig. 5a–d and Supplementary Fig. 6a). In parallel, we performed LDA and ELDA on *OSMR* KD BTSC73 and BTSC147 in which stable or transient KD of *OSMR* was achieved by CRISPR or *OSMR* siRNAs, respectively. Both transient or stable KD of *OSMR* in BTSCs resulted in a significant decrease in sphere numbers as well as the mean sphere size in response to IR (Fig. 5e–h and Supplementary Fig. 6b–g). A significant decrease in cell viability was also observed in irradiated *OSMR* CRISPR BTSCs compared to irradiated control BTSCs as assessed by alamarBlue assay (Fig. 5i, j). Similar to BTSCs, we found that addition of OSM to OSMR-over-expressing primary CGN cultures conferred resistance to DNA damage-induced cell death in response to camptothecin (CPT) (Supplementary Fig. 7a). Together, these results established that OSM/OSMR signalling pathway confers resistance to DNA damage-induced cell death in different model systems.

Since IR triggers the production of ROS which plays an important role in promoting cell death[34], we examined the levels of ROS in *OSMR* CRISPR and control BTSC73 following IR treatment. Using H2DCFDA and MitoSOX flow cytometry assays, we assessed ROS levels in *OSMR* CRISPR and control BTSCs following 24 h of irradiation. Our data revealed an increase in ROS production in control irradiated BTSCs compared to non-irradiated BTSCs. Importantly, the ROS levels were significantly elevated in irradiated *OSMR* CRISPR BTSCs relative to irradiated control BTSCs (Fig. 6a, b). Consequently, excess ROS production was followed by an increase in BTSC death, whereby 40% of irradiated *OSMR* CRISPR BTSCs were positive for annexin V compared to only 20% in the irradiated control BTSCs (Fig. 6c).

**OSMR suppression improves glioblastoma response to IR.** To investigate the functional relevance of these findings to

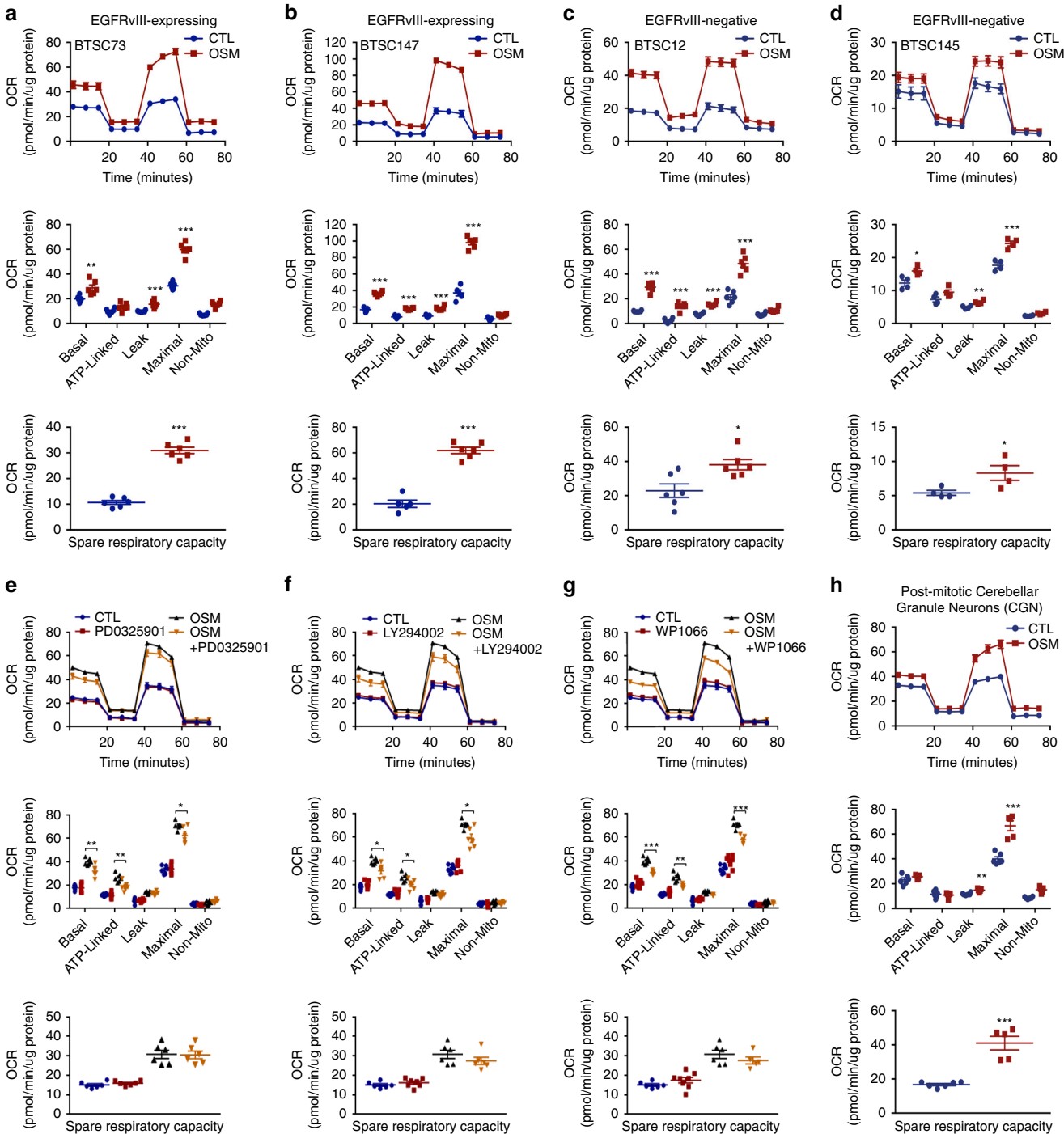

**Fig. 4 The ligand OSM regulates mitochondrial OXPHOS. a**, **b** Bioenergetic analysis using a Seahorse XFe96 Bioenergetic Flux Analyzer in the absence and presence of OSM was performed in EGFRvIII-expressing BTSCs. BTSC73 (**a**): ***$p < 0.0001$ for each pairwise comparison except: **$p_{Basal} = 0.0045$, ***$p_{Leak} = 0.0006$; Unpaired two-tailed $t$-test, $n = 6$; BTSC147 (**b**): ***$p < 0.0001$ for each pairwise comparison; Unpaired two-tailed $t$-test, $n \geq 5$. **c**, **d** Bioenergetic analysis in the absence and presence of OSM was performed in BTSCs lacking the EGFRvIII mutation. BTSC12 (**c**): ***$p < 0.0001$ for each pairwise comparison except: *$p_{SRC} = 0.0124$; Unpaired two-tailed $t$-test, $n = 6$; BTSC145 (**d**): *$p_{Basal} = 0.0179$, *$p_{SRC} = 0.0435$, **$p_{Leak} = 0.0056$, ***$p_{Maximal} = 0.0008$; Unpaired two-tailed $t$-test, $n = 4$. **e**–**g** BTSC73 were subjected to bioenergetic analysis in the absence and presence of 10 ng/mL of OSM and either of pharmacological inhibitors 10 μM PD0325901, 10 μM LY294002, or 20 μM WP1066. PD0325901 (**e**): *$p_{Maximal} = 0.0303$, **$p_{ATP-Linked} = 0.0088$, **$p_{Basal} = 0.0071$; Unpaired two-tailed $t$-test, $n = 6$; LY294002 (**f**): *$p_{ATP-Linked} = 0.0322$, *$p_{Basal} = 0.0104$, *$p_{Maximal} = 0.0101$; Unpaired two-tailed $t$-test, $n \geq 6$; WP1066 (**g**): **$p_{ATP-Linked} = 0.0029$, ***$p_{Basal} = 0.0001$, ***$p_{Maximal} = 0.0002$; Unpaired two-tailed $t$-test $n \geq 5$. **h** OSMR-overexpressing primary CGN cultures were subjected to bioenergetic analysis in the absence or presence of OSM. **$p_{Leak} = 0.0014$, ***$p_{Maximal} = 0.0001$, ***$p_{SRC} < 0.0001$; Unpaired two-tailed $t$-test, $n \geq 5$. Data are presented as the mean ± SEM. $n$ represents an independent biological sample.

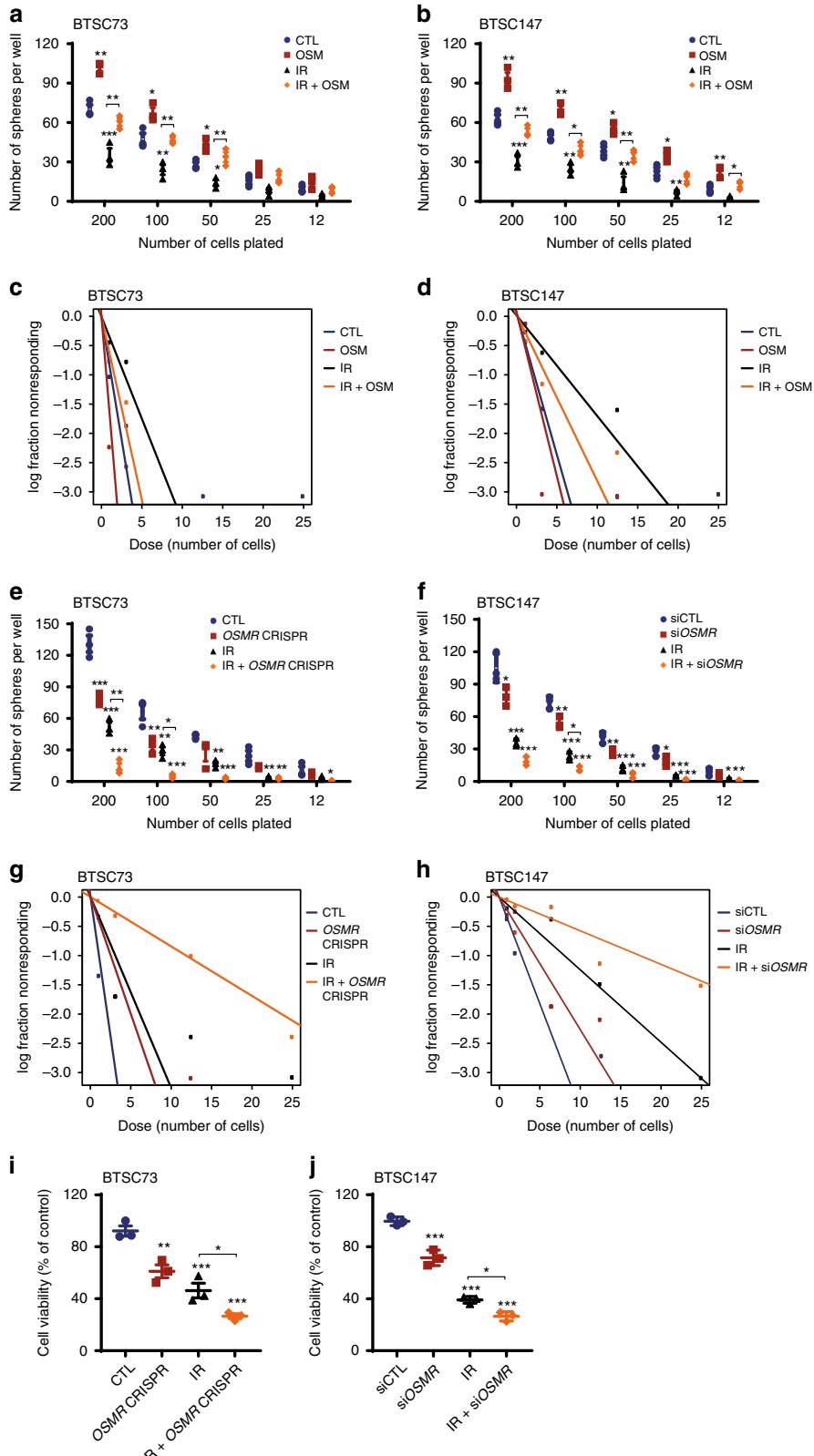

glioblastoma in vivo, we performed intracranial xenografts of *OSMR* CRISPR or control BTSC73 in immunodeficient SCID mice and assessed whether therapy with IR in mice receiving *OSMR* CRISPR improves the lifespan of the animals. Mice receiving *OSMR* CRISPR or control BTSCs were subjected to IR, and live imaging of luciferase activity in the brains was performed

using the In Vivo Imaging System (IVIS) to trace BTSCs and tumor volume (Fig. 7a). At 17 days following surgery, mice receiving control BTSC73 formed malignant brain tumors in the absence of IR and were at endpoints as assessed by major weight loss and neurological signs (Fig. 7b–e). KD of *OSMR* by 50% or exposure to 4 Gy of IR delayed tumorigenesis, whereby mice

**Fig. 5 OSM/OSMR confers resistance of BTSCs to IR. a, b** LDA was performed following 4 Gy of IR in the absence or presence of OSM. BTSC73 (**a**): 200 cells (**$p_{CTL \ vs. \ OSM}$ = 0.0011, **$p_{IR \ vs. \ IR + OSM}$ = 0.0054, ***$p_{CTL \ vs. \ IR}$ = 0.0007), 100 cells (*$p_{CTL \ vs. \ OSM}$ = 0.0185, **$p_{CTL \ vs. \ IR}$ = 0.0078, **$p_{IR \ vs. \ IR + OSM}$ = 0.0093), 50 cells (*$p_{CTL \ vs. \ OSM}$ = 0.0246, *$p_{CTL \ vs. \ IR}$ = 0.0389, **$p_{IR \ vs. \ IR + OSM}$ = 0.0054); BTSC147 (**b**): 200 cells (**$p_{CTL \ vs. \ OSM}$ = 0.0012, **$p_{IR \ vs. \ IR + OSM}$ = 0.0088, ***$p_{CTL \ vs. \ IR}$ = 0.0009), 100 cells (*$p_{IR \ vs. \ IR + OSM}$ = 0.0236, **$p_{CTL \ vs. \ OSM}$ = 0.0026, **$p_{CTL \ vs. \ IR}$ = 0.0012), 50 cells: (*$p_{CTL \ vs. \ OSM}$ = 0.0301, **$p_{CTL \ vs. \ IR}$ = 0.0039, **$p_{IR \ vs. \ IR + OSM}$ = 0.0087), 25 cells (*$p_{CTL \ vs. \ OSM}$ = 0.0382, **$p_{CTL \ vs. \ IR}$ = 0.0044), 12 cells (**$p_{IR \ vs. \ IR + OSM}$ = 0.0207, **$p_{CTL \ vs. \ OSM}$ = 0.0044). **c, d** ELDA was performed following 4 Gy of IR in the absence or presence of OSM in either BTSC73 (**c**) or BTSC147 (**d**). **e, f** LDA were performed following 4 Gy of IR in OSMR KD and control BTSCs. ***$p$ < 0.0001 except: BTSC73 (**e**): 200 cells (**$p_{IR \ vs. \ IR + OSMR \ CRISPR}$ = 0.0030, ***$p_{CTL \ vs. \ OSMR \ CRISPR}$ = 0.0004), 100 cells (*$p_{IR \ vs. \ IR + OSMR \ CRISPR}$ = 0.0286, **$p_{CTL \ vs. \ IR}$ = 0.0021, **$p_{CTL \ vs. \ OSMR \ CRISPR}$ = 0.0052), 50 cells (**$p_{CTL \ vs. \ IR}$ = 0.0061, ***$p_{CTL \ vs. \ IR + OSMR \ CRISPR}$ = 0.0004), 25 cells (**$p_{CTL \ vs. \ IR}$ = 0.0018, **$p_{CTL \ vs. \ OSMR \ CRISPR + IR}$ = 0.0016), 12 cells (**$p_{CTL \ vs. \ IR + OSMR \ CRISPR}$ = 0.0404); BTSC147 (**f**): 200 cells (*$p_{siCTL \ vs. \ siOSMR}$ = 0.0184), 100 cells (*$p_{IR \ vs. \ IR + siOSMR}$ = 0.0414, **$p_{siCTL \ vs. \ siOSMR}$ = 0.0035), 50 cells (**$p_{siCTL \ vs. \ siOSMR}$ = 0.0050), 25 cells (*$p_{siCTL \ vs. \ siOSMR}$ = 0.0264, ***$p_{CTL \ vs. \ IR}$ = 0.0001), 12 cells (*$p_{siCTL \ vs. \ IR}$ = 0.0194, **$p_{siCTL \ vs. \ IR + siOSMR}$ = 0.0070). **g, h** ELDA was performed following 4 Gy of IR in OSMR CRISPR vs. control BTSC73 (**g**) or in siOSMR vs. siCTL BTSC147 (**h**). **i, j** Cell viability was measured by alamarBlue assay following 4 Gy of IR in OSMR KD and control BTSCs. ***$p$ < 0.0001 except: BTSC73 (**i**): *$p_{IR \ vs. \ IR + OSMR \ CRISPR}$ = 0.0490, **$p_{CTL \ vs. \ OSMR \ CRISPR}$ = 0.0044, ***$p_{CTL \ vs. \ IR}$ = 0.0003; BTSC147 (**j**): *$p_{IR \ vs. \ IR + siOSMR}$ = 0.0213. Data are presented as the mean ± SEM, $n$ = 3 independent biological cell cultures, one-way ANOVA followed by Tukey's test for multiple comparisons.

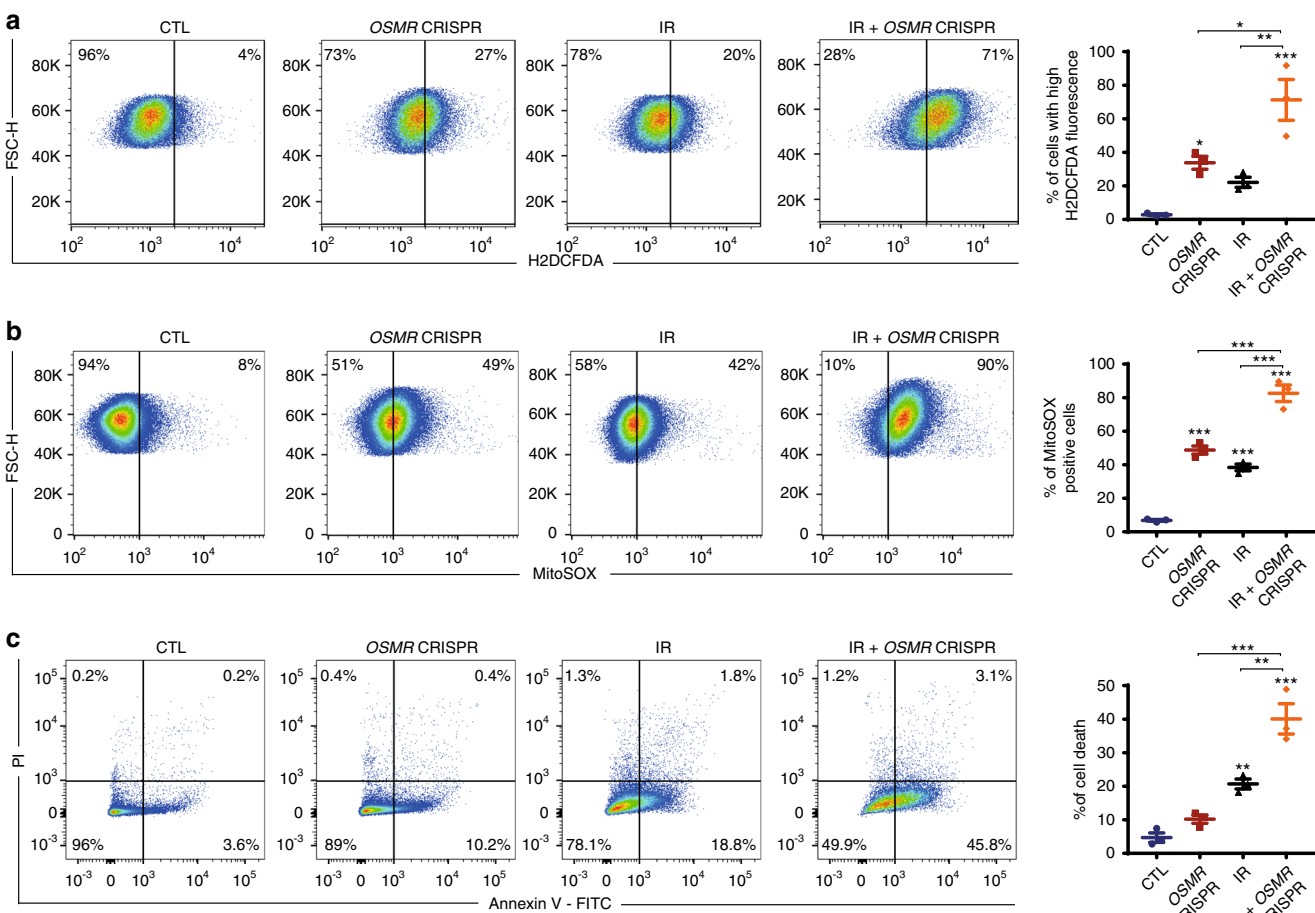

**Fig. 6 OSMR CRISPR induces ROS generation and promotes apoptosis in response to IR. a–c** OSMR CRISPR and control BTSC73 were subjected to IR (8 Gy). ROS generation was analyzed by flow cytometry following 24 h after IR using H2DCFDA (**a**): *$p_{CTL \ vs. \ OSMR \ CRISPR}$ = 0.0416, *$p_{OSMR \ CRISPR \ vs. \ IR + OSMR \ CRISPR}$ = 0.0158, **$p_{IR \ vs. \ IR + OSMR \ CRISPR}$ = 0.0033, ***$p_{CTL \ vs. \ IR + OSMR \ CRISPR}$ = 0.0004; One-way ANOVA followed by Tukey's test for multiple comparisons, $n$ = 3 independent biological samples. Mitochondrial superoxide abundance was assessed by flow cytometry 24 h after IR using MitoSOX (**b**): ***$p$ < 0.0001 for each pairwise comparison except: ***$p_{CTL \ vs. \ IR}$ = 0.0003, ***$p_{OSMR \ CRISPR \ vs. \ IR + OSMR \ CRISPR}$ = 0.0002; One-way ANOVA followed by Tukey's test for multiple comparisons, $n$ = 3 independent biological samples. Apoptosis analysis was performed by flow cytometry 48 h after IR by annexin V and PI double staining (**c**). The percentage of cell death (annexin V positive cells) is presented in the histogram (right panel), **$p_{CTL \ vs. \ IR}$ = 0.0094, **$p_{IR \ vs. \ IR + OSMR \ CRISPR}$ = 0.0029, ***$p_{OSMR \ CRISPR \ vs. \ IR + OSMR \ CRISPR}$ = 0.0002, ***$p_{CTL \ vs. \ IR + OSMR \ CRISPR}$ < 0.0001; One-way ANOVA followed by Tukey's test for multiple comparisons, $n$ = 3 independent biological samples. Data are presented as the mean ± SEM. Representative scatter plots (left) and histograms (right) of flow cytometry analyses are shown.

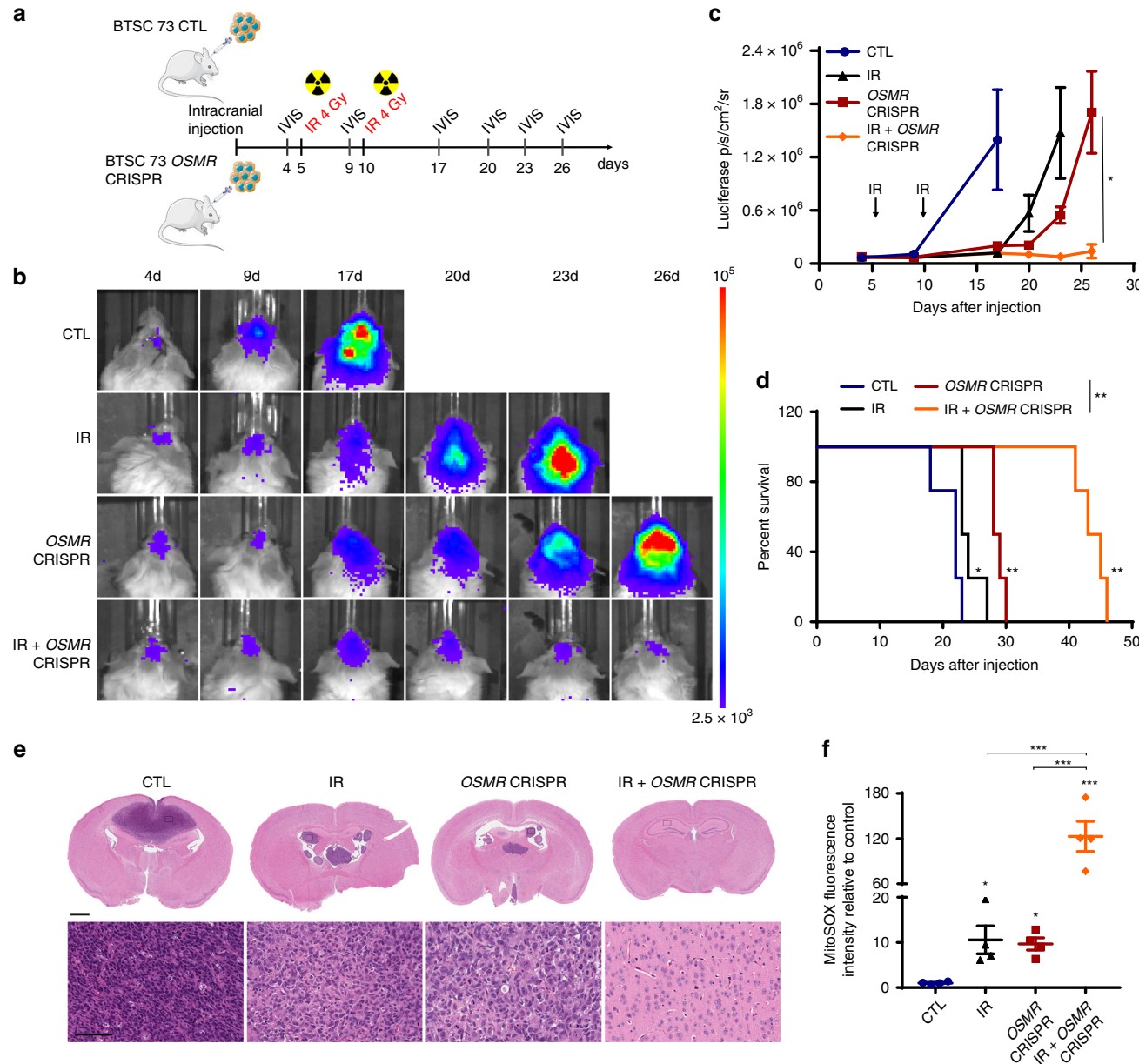

**Fig. 7 Suppression of OSMR improves glioblastoma response to therapy. a** Schematic diagram of the experimental procedure in which *OSMR* CRISPR and control BTSC73 (3 × 10⁵ cells per brain) were intracranially injected into randomized Fox Chase SCID mice and then treated with or without 4 Gy of IR. **b** Representative bioluminescence real-time images tracing BTSCs and tumor growth are shown. **c** Intensities of luciferase signal were quantified at different time points using Xenogen IVIS software. *$p_{OSMR}$ CRISPR vs. IR + OSMR CRISPR, day 26 = 0.0289; Unpaired two-tailed *t*-test, n = 4 mice. **d** Kaplan–Meier survival plot was graphed to evaluate mice lifespan in each group, mice were collected at end stage. *$p_{CTL}$ vs. IR = 0.0311, **$p_{CTL}$ vs. OSMR CRISPR = 0.0069, **$p_{CTL}$ vs. IR + OSMR CRISPR = 0.0069, **$p_{OSMR}$ CRISPR vs. IR + OSMR CRISPR = 0.0067; Two-sided log-rank test, n = 4 mice. **e** Coronal sections of mouse brains were stained with hematoxylin and eosin on day 22 after injection. Representative images of 3 different tumor sections are shown. Scale bar = 1 mm. Inset scale bar = 0.1 mm. **f** Tumor sections from *OSMR* CRISPR and control BTSC73 treated with or without IR (4 Gy) were stained with MitoSOX. The fluorescent intensity was quantified using ImageJ software. ***$p < 0.0001$ for each pairwise comparison except: *$p_{CTL}$ vs. OSMR CRISPR = 0.0215, *$p_{CTL}$ vs. IR = 0.0129; One-way ANOVA followed by Tukey's test for multiple comparisons, n = 4 different tumor sections. Data are presented as the mean ± SEM.

exposed to IR or mice receiving *OSMR* KD BTSCs were at endpoint at 23 and 26 days, respectively. Strikingly, mice receiving a combination of *OSMR* KD BTSCs and IR survived past 40 days (Fig. 7b–e). To determine whether *OSMR* KD affects ROS generation following exposure to IR in vivo, mitochondrial superoxide levels were assessed via staining of the patient-derived tumor sections with MitoSOX. KD of *OSMR* induced a significant increase in ROS levels compared to control as detected via analysis of MitoSOX fluorescence signal intensity. Importantly,

exposure to IR combined with KD of *OSMR* significantly enhanced ROS levels whereby we observed a 120-fold increase in the fluorescence signal intensity compared to only ~10-fold in either irradiated control tumors or *OSMR* KD tumors in the absence of IR (Fig. 7f and Supplementary Fig. 7b). In conclusion, our results established that KD of *OSMR* increases cellular ROS levels, promotes BTSC death, sensitizes the response of BTSCs and brain tumors to IR, and most importantly extends animal lifespan.

## Discussion

In the present study, we report the discovery of a mitochondrial OSMR that functions to promote OXPHOS and confers resistance of glioblastoma tumors to IR. Beginning with unbiased IP-LC-MS/MS screening, followed by PLA, co-IP experiments, high resolution confocal imaging and FRAP experiments, we established that OSMR is targeted to the mitochondria via PAM complex and directly interacts with different components of mitochondrial complex I. Via Seahorse bioenergetic analyses and assessing the enzymatic activity of different ETC complexes, we established that OSMR regulates OCR and mitochondrial respiration. Importantly, we found that KD of *OSMR* in glioma stem cells generates excess ROS and induces cell death in vitro and in vivo. More importantly, we established that loss of OSMR sensitizes the response of glioblastoma tumors to IR therapy. Our data suggest that OSMR targeting in combination with IR provides a promising therapeutic approach for better treatment of glioblastoma tumors.

*OSMR* is a direct STAT3 target gene. At the same time, OSMR forms a co-receptor complex with EGFRvIII which together induce the phosphorylation of STAT3[20]. Members of STAT family, including STAT3, have been reported in the mitochondria[29,35]. Interestingly, both of EGFRvIII and wild type EGFR are present in the mitochondria whereby they protect cancer cells from apoptotic cell death[36,37]. Our discovery of a mitochondrial OSMR led to the question of whether OSMR regulation of cellular respiration depends on EGFRvIII and STAT3. Inhibition of STAT3 phosphorylation by the pharmacological inhibitor, WP1066, in OSM-treated BTSCs led to a modest decrease in OCR, suggesting that the profound impact of OSM on cellular respiration overrides the STAT3 effect. In parallel, we examined the contribution of EGFRvIII to OSM-mediated regulation of respiration. We found that similar to BTSCs harboring the EGFRvIII mutation, BTSCs lacking the EGFRvIII mutation, significantly responded to OSM and exhibited upregulated OCR. Most importantly, using wild type tissue and post-mitotic primary CGN cultures, we established that ectopic expression of OSMR together with OSM addition are sufficient to induce significant upregulation of mitochondrial respiration. Taken together, we report a unique function for OSMR in respiration independent of its role in cell proliferation or EGFRvIII/STAT3 regulation.

Mitochondria undergo constant cycles of fission and fusion to regulate their shape and function[38–40]. Mitochondrial (mt) EGFR and EGFRvIII have been reported to alter glioblastoma metabolism and promote tumor progression via interacting with the pyruvate dehydrogenase kinase[41]. Interestingly, mtEGFR cooperates with the mitochondrial fusion protein, mitofusin-1, and regulates mitochondrial dynamics[42]. Whether OSMR impacts mitochondrial respiration via interaction with the mitochondrial fission and fusion proteins is an exciting area of research to be explored.

Complex I is the major source of mitochondrial ROS[25,43]. We established that OSMR directly interacts with NDUFS1/NDUFS2 of complex I in the mitochondria of patient derived glioma stem cells. Loss of NDUFS1 function is shown to impair OCR and lead to the accumulation of ROS[25,44,45]. Furthermore, *NDUFS1/2* mutations are associated with complex I deficiency that is presented in different OXPHOS disorders[44–46]. Similar to *NDUFS1*, *OSMR* deletion impairs OCR and leads to excess ROS suggesting that loss of *OSMR* phenocopies *NDUFS1* KD effects. Our observations suggest a model whereby OSMR/NDUFS1 complex function in the same pathway to regulate mitochondrial respiration and buffer excess ROS.

The function of the mitochondrial respiratory chain involves the organization of the enzyme complexes into respirasomes or supercomplexes that are structurally interdependent[43,47–49]. Therefore, defects in a single ETC component often produce combined enzyme deficiencies in patients[50–52]. The formation of supercomplexes has been reported to be severely reduced in mutated *NDUFS1* patient[53]. Furthermore, *NDUFS1* KD in primary cortical neurons was shown to decrease the assembly of complex I into supercomplexes[25]. We found that although OSMR interacts with complex I, loss of OSMR impairs the enzymatic activities of all four complexes in the ETC. Our data support a model whereby OSMR may contribute to the assembly of supercomplex via a cross talk with NDUFS1.

Radioresistance has emerged as one of the major obstacles in glioblastoma therapy. BTSCs have been shown to be more radioresistant than their non-stem cell counterparts in glioblastoma[7,31]. Interestingly, an increase in cellular respiration and SRC confers resistance to IR therapy[13,54–56], and inhibitors of ETC have been described as selective anti-cancer agents[57–61]. For example, the ATP synthase inhibitor, Gboxin, has been recently discovered to target glioblastoma cells and reduce tumorigenesis in vitro and in vivo[62]. Furthermore, the complex I inhibitor, metformin, in combination with TMZ and IR is being investigated as a therapeutic avenue for glioblastoma patients[63]. Thus, our findings support that the use of OSMR and complex I inhibitors in combination with IR may provide a promising approach in better treatment of glioblastoma patients. Importantly, OSMR serves as an attractive therapeutic target for multiple reasons. First, OSMR expression level is highly upregulated in high grade glioma and upregulation of OSMR is an important predictor of poor patient survival[21]. Second, OSMR is highly expressed in glioma stem cells[20]. Third, *OSMR* knockout mice are viable, healthy and fertile[64], thus OSMR inhibitors are deemed to pose minimum side effects.

In conclusion, our study has uncovered an important mechanism whereby cytokine signalling alters BTSC metabolism and confers glioblastoma resistance to therapy. OSM/OSMR targeted therapies are promising in eliminating IR-resistant BTSCs in the tumor mass, impairing mitochondrial function, and improving response to IR.

## Methods

**Patient-derived BTSC cultures.** The human BTSC line 112, 145, and 172 were generously provided by Dr. Keith Ligon at Harvard Medical School. BTSC lines were generated following surgery with informed consent of adult glioblastoma patients following the BWH/Partners IRB protocol for use of excess/discarded tissue at Harvard University. BTSC12, 73, and 147 were provided by Dr. Samuel Weiss at the University of Calgary. Cells were characterized for major mutations in glioblastoma including EGFRvIII, p53, PTEN, and IDH1 status[20]. BTSCs 73, 147, 112, and 172 that naturally harbor EGFRvIII mutations, and BTSCs 12 and 145 that do not harbor the mutation, were used in this study. Prior to use, BTSCs were recovered from cryopreservation in 10% dimethyl sulfoxide and cultured in Nunc ultra-low attachment flasks as neurospheres in NeuroCult NS-A medium (Stemcell Technologies, #05750) supplemented with 100 U/mL penicillin, 100 μg/mL streptomycin (Sigma Aldrich, #P4333), heparin (2 μg/mL, Stemcell Technologies, #07980), human EGF (20 ng/mL, Miltenyi Biotec, #130-093-825), and human FGF (10 ng/mL, Miltenyi Biotec, #130-093-838). All cell lines were tested negative for mycoplasma.

**Generation of transgenic BTSCs.** We employed 4 different approaches to KD *OSMR* in patient-derived human BTSCs.

First, genetic deletion of *OSMR* was achieved using CRISPR[65]. Briefly, two gRNAs were designed using off-spotter software to delete exon 5-7 resulting in a 2.8 kb deletion of *OSMR* gene. gRNA-1 and -2 were cloned into pL-CRISPR.EFS.GFP and pL-CRISPR.EFS.tRFP, respectively. 5 ug of each construct were nucleofected into BTSC73 using an AMAXA nucleofector 2b device (Lonza, #AAB-1001). The GFP and RFP positive cells were then sorted two days post-electroporation and plated clonally using FACSAria Fusion (Supplementary Fig. 2a). Genomic DNA was isolated from each clone and screened for *OSMR* deletion via PCR using specific internal and external primers around the site of the deletion. This led to the identification of monoallelic deletion, biallelic deletion and non-deletion clones. Since the biallelic deletion of *OSMR* impaired BTSC growth or led to cell death, we selected monoallelic deletion clones for follow up analysis.

*OSMR* mRNA and protein levels were analyzed by RT-qPCR and WB, respectively, to assess KD levels (Supplementary Fig. 2b, c). The following gRNAs and screening primers were used for CRISPR/Cas9 system:

gRNA-1-*OSMR*-Fwd: caccgAGTACAATGAAGAGATTACG
gRNA-1-*OSMR*-Rev: aaacCGTAATCTCTTCATTGTACTc
gRNA-2-*OSMR*-Fwd: caccgAGCACAGCGTTAGTGGCCACGG
gRNA-2-*OSMR*-Rev: aaacCCGTGGCCACTAACGTGTCTc
*OSMR* External-Fwd: AGAAGGACACATACACAGGGAA
*OSMR* External-Rev: GCGTGCATCCATGAGGAGAA
*OSMR* Internal-Fwd: AGCATCTCCTTCCCTTGCAC
*OSMR* Internal-Rev: AGCATCTCCTTCCCTTGCAC.

Second, the transgenic *OSMR* KD BTSCs were generated via lentivirus carrying two different *OSMR* shRNA plasmids[20]. *OSMR* KD BTSC73 lines were established by antibiotic selection (0.5 μg/mL puromycin). As control, a lentivirus carrying a non-targeting construct was used.

Third, we generated an inducible *OSMR* KD line using CRISPRi-dCas9. For the generation of CRISPRi-dCas9 expressing lines, 5 μg of pAAVS1-NDi-CRISPRi construct (Addgene, #73498) and 2 μg of AAVS1 TALEN pair constructs (Addgene, #59025 and #59026) were nucleofected into BTSC73. 100 ng/mL geneticin was added to select transfected cells. Fluorescence-activated cell sorting (FACS) was performed on BTSC73 that were induced with 2 μM doxycycline for 96 h to obtain a homogeneous CRISPRi mCherry-expressing BTSCs. Using the MIT CRISPR design tool, 2 gRNAs were designed for *OSMR* within 250 bp of the transcription start site. gRNA oligos were cloned into the pgRNA-CKB vector using BsmBI restriction enzyme ligation[66], and 5 μg was nucleofected into CRISPRi mCherry-expressing BTSC73. Cells were sorted for mCherry, followed by selection with 10 μg/mL blasticidin (Supplementary Fig. 2d, e). *OSMR* mRNA and protein expression levels were analyzed by RT-qPCR and immunoblotting following doxycycline addition (Supplementary Fig. 2f, g).

The following gRNAs were used for CRISPRi-dCas9 system:

gRNA-1-*OSMR*-Fwd: TTGGGGAGCCGGGCCGAGTCCTCGG
gRNA-1-*OSMR*-Rev: AAACCCGAGGACTCGGCCCCGGCTC
gRNA-2-*OSMR*-Fwd: TTGGGCCCGGGCCTGCCTACCTGGT
gRNA-2-*OSMR*-Rev: AAACACCAGGTAGGCAGGCCGGGC.

Fourth, we conducted transient KD of *OSMR* using siRNA approach. ON TARGET-plus SMART pool human *OSMR* siRNA (Dharmacon, #L-008050-00-0005), and ON TARGET-plus non-targeting pool (Dharmacon, #D-001810-10-05) were used. siRNA (100 nM) were nucleofected into BTSCs (10^6 cells) and cultured in BTSC media at 37 °C in a humidified atmosphere of 5% $CO_2$.

To overexpress *OSMR*, lentiviruses containing pUNO1 RFP plasmid bearing the coding sequence of human *OSMR* (InvivoGen, #puno1-hosmr) were used to generate *OSMR* over-expressing BTSCs. *OSMR* over-expressing BTSCs were established by FACS of the RFP-positive live cells followed by antibiotic selection (0.5 μg/mL puromycin).

To generate GFP-tagged OSMR expressing BTSCs, lentiviruses containing pCMV6-*OSMR*-GFP mammalian vector (OriGene Technologies, #RG216943) were used. GFP-OSMR over-expressing BTSCs were then maintained by antibiotic selection (2 μg/mL geneticin).

**Primary CGNs culture.** Primary CGNs were isolated from 6–7 days old mouse pups by mechanical and enzymatic dissociation[67]. Pups were euthanized according to the McGill animal use and care committee guidelines. Briefly, neurons were isolated from freshly dissected cerebella by using trypsin (Sigma Aldrich, #85450 C), treated with DNase (Sigma-Aldrich, #11284932001), and plated at a density of $0.7 \times 10^6$ cells/cm² on cell culture plates coated with poly D-lysine (VWR, #89134-858) and cultured in Eagle's minimal essential media (Sigma Aldrich, #M-2279) supplemented with 1.125 g/L D-glucose (Sigma-Aldrich, #G-7528), 10% of heat-inactivated dialyzed Fetal Bovine Serum (FBS) (Sigma Aldrich, #F0392), 2 mM of L-Glutamine (Gibco, #25030081), 0.1 mg/mL gentamycin (Sigma-Aldrich, #G-1397), and 20 mM of potassium chloride (VWR, #CABDH9258). Neurons were maintained at 37 °C in a 5% CO2 incubator for up to 7 days. 24 h following plating, 10 μM cytosine-β-arabino furanoside (Sigma-Aldrich, #C-1768) was added to reduce glial contamination[67]. To overexpress OSMR, CGNs were infected with adenovirus bearing pAdenoG vector that contains the coding sequence of mouse *OSMR* (abm, #218317 A) at the time of plating.

**Cell fractionation and isolation of mitochondria.** Intact mitochondria were isolated and purified using Qproteome Mitochondria Isolation Kit (Qiagen, #37612) from liver and brain of 8-week-old C57BL/6J mice and BTSCs according to the manufacturer's instructions. Briefly, $2 \times 10^7$ BTSCs were washed with 0.9% sodium chloride solution and resuspended in lysis buffer. Tissues were homogenized using the TissueRuptor rotor-stator homogenizer. Tissues and cells were incubated at 4 °C for 10 min on a shaker and were subjected to centrifugation (1000 $g$, 10 min, 4 °C). The supernatants containing cytosolic proteins were carefully removed. The pellets were resuspended in disruption buffer and were subjected to centrifugation (1000 $g$, 10 min, 4 °C). The pellets containing nuclei were resuspended in RIPA lysis buffer and the supernatants were centrifuged at 6000 $g$ (10 min, 4 °C) to obtain the mitochondrial fraction. Protein concentration of mitochondrial, cytosolic, and nuclei lysates was determined using the Bradford assay (Bio-Rad, #5000006).

**Proteinase K sensitivity.** Proteinase K sensitivity experiments were conducted with the incubation of the intact mitochondria with proteinase K at 0.5 μg/mL in the presence or absence of 1% Triton X-100. After 7 min of incubation at room temperature, reactions were stopped by adding 2 mM phenylmethylsulfonyl fluoride. Samples were analyzed by immunoblotting.

**Protein immunoprecipitation.** Immunoprecipitations were performed from WCL or pure mitochondrial fractions of BTSCs, EGFRvIII-expressing mouse astrocytes or primary CGNs. Cells were lysed for 30 min on ice in lysis buffer (50 mM Tris pH 7.5, 150 mM NaCl, 2 mM $MgCl_2$, 0.5 mM EDTA, 0.5% Triton X-100, protease inhibitor cocktail). Lysates were cleared by centrifugation (14,800 $g$, 20 min, 4 °C) and subsequently incubated with either anti-OSMR antibody (Abnova, #H00009180-B01P) or mouse IgG (Millipore, #12-371) as a control. Primary antibody incubations were carried out overnight at 4 °C, followed by a 1 h incubation at room temperature with Dynabeads Protein G magnetic beads (Thermo Fisher Scientific, #10003D). As an additional control, beads were incubated with WCL or mitochondrial fractions without primary antibodies. Beads were washed three times with lysis buffer and eluted by boiling in SDS sample buffer. Immunoprecipitates were analyzed by immunoblotting using the indicated antibodies. Large scale IPs were performed on EGFRvIII-expressing astrocytes using the same protocol and samples were submitted for LC-MS/MS analysis at the Institute for research in Immunology and Cancer, Montréal, Canada.

**Immunoblotting and antibodies.** Total protein was harvested in RIPA lysis buffer containing protease and phosphatase inhibitors (Thermo Fisher Scientific, #A32959). Protein concentration was determined by Bradford assay (Bio-Rad), after which samples were subjected to SDS-PAGE and electroblotted onto Immobilon-P membrane (Millipore). Membranes were blocked in 5% nonfat milk or 5% bovine serum albumin (BSA) in TBST, before sequential probing with primary antibodies and HRP-conjugated secondary antibodies in blocking solution[68]. Target proteins were visualized by ECL (Biorad) using ChemiDoc Imaging System (Biorad). Uncropped scans are shown in Supplementary Fig. 8. The following antibodies were used: OSMR (1:100, Santa Cruz, #271695), mtHSP70 (1:1000, Invitrogen, #MA3-028), TIM44 (1:500, Abcam, #244466), TOM20 (1:1000, Cell Signalling, #42406), H3K4me3 (1:1000, Abcam, #8580), BCL2 (1:1000, Cell Signalling, #15071), prohibitin (1:1000, Cell Signalling, #2426), NDUFS1 (1:3000, Abcam, #169540), NDUFS2 (1:4000, Abcam, #110249), α-tubulin (1:5000, Abcam, #4074), Na+/K+ ATPase (1:1000, Abcam, #58475), calnexin (1:1000, Abcam, #22595), phospho-Akt (Ser473) (1:1000, Cell Signalling, #4060), phospho-p44/42 MAPK (Thr202/Tyr204) (1:1000, Cell Signalling, #9101) and phospho-STAT3 (Tyr705) (1:1000, Cell Signalling, #9138).

**Ionizing radiation experiments.** For measurement of ROS generation and cell death, BTSCs were dissociated to single cell suspension using Accumax (Innovative Cell Technologies, #AM105). 10^6 BTSCs were plated and irradiated with either 4 or 8 Gy using the X-Ray Irradiation System (Faxitron MultiRad 225).

**Limiting dilution assay and extreme limiting dilution assay.** For LDA, *OSMR* KD or 10 ng/mL OSM-treated BTSCs were dissociated to single cell suspension using Accumax, counted and plated in 96-well plate at different densities ranging from 200 to 12 cells per well in triplicates. Spheres were counted 7 days after plating. Spheres were visualized using 10X objective on an Olympus IX83 microscope with an Olympus DP80 camera and sphere sizes (diameters) were measured using Olympus cellSens Software.

For ELDA experiments, decreasing numbers of BTSCs per well (dose: 25, 12, 6, 3 and 1) were plated in a 96-well plate with a minimum of 12 wells/dose. Seven days after plating, the presence of spheres in each well was recorded and analysis was performed using software available at http://bioinf.wehi.edu.au/software/elda/[33].

**Cell viability and cell death assessment.** BTSCs were dissociated to single cell suspension using Accumax. Cells were seeded at a density of 200 cells/well, in a 96-well plate and irradiated with 4 Gy using the X-Ray Irradiation System (Faxitron MultiRad 225). Cell viability was evaluated 7 days post-plating using alamarBlue (Thermo Fisher Scientific, #DAL1100) according to the manufacturer's protocol. Briefly, 10% of resazurin was added per well and cells were incubated for 4 h at 37 °C. Fluorescence was read using a fluorescence excitation wavelength of 560 nm and a emission of 590 nm.

For BTSCs, cell death was determined using TACS annexin V-FITC apoptosis detection kit (R&D systems 4830-01-K) according to the manufacturer's protocol. Briefly, 10^6 cells were seeded in a 25 cm² flask. Following 48 h of seeding, cells were chemically dissociated using Accumax, washed with cold PBS, and co-stained with TACS annexin V-FITC and propidium iodide (PI) (R&D, #4830-01-K) following the manufacturer's instructions. The fluorescence was analyzed by flow cytometry (BD FACS CantoII). Data were analyzed using the FlowJo software. Both early apoptotic (annexin V-positive, PI-negative) and late apoptotic (annexin V-positive and PI-positive) cells were included in the cell death plots.

To assess cell death in CGN, neurons were treated with 10 μM CPT following 3 h of incubation with 100 ng/mL of OSM. Neurons were stained with DAPI following 16 h of treatment with CPT. Pyknotic nuclei were counted using ImageJ.

Percent cell death was calculated as the ratio of pyknotic nuclei to the total number of cells in the field. A minimum of 10 randomly selected fields were imaged and counted for each biological replicate. Images were acquired using a 60X objective on an Olympus IX83 microscope with an X-Cite 120 LED from Lumen Dynamics and an Olympus DP80 camera.

**Measurement of electron transport chain complex activities.** The activities of ETC complexes I, II and IV were measured spectrophotometrically using the SmartSpec Plus spectrophotometer (Biorad) as previously described[69]. All assays were performed using potassium phosphate buffer as assay buffer at 50 mM for complex I, and 25 mM for complex II and IV. Complex I (NADH-ubiquinone oxidoreductase) activity was determined in a reaction mixture containing assay buffer, 100 µM NADH, 60 µM ubiquinone-1, 300 µM potassium cyanide (KCN), and 3 mg/mL BSA. The reactions were initiated by the addition of 20 µg of mitochondrial proteins and the change in absorbance due to NADH oxidation in the presence of ubiquinone-1 was monitored spectrophotometrically at 340 nm. Parallel reactions were performed in the presence of 10 µM rotenone. The specific activity of complex I is calculated by subtracting the rotenone-resistant activity (with rotenone) from the total complex I activity (without rotenone). To measure complex II (succinate: ubiquinone oxidoreductase) activity, a reaction mixture containing assay buffer, 20 mM succinate, 80 µM DCPIP (2,6-dichlorophenol-indophenol), 300 µM KCN and 1 mg/mL BSA was incubated with 20 µg of WCL for 10 min at 37 °C. Then, the reaction was initiated by adding 50 µM decylubi-quinone and the change in absorbance due to the reduction of DCPIP was mon-itored spectrophotometrically at 600 nm. Parallel reactions were performed in the presence of 10 mM malonate. The specific activity of complex II was obtained by subtracting malonate-resistant activity (with malonate) from the total complex II activity (without malonate). Complex IV (cytochrome c oxidase) activity was determined in a reaction mixture that consisted of assay buffer and 50 µM of reduced cytochrome c. The reaction was initiated by the addition of 20 µg of WCL proteins and the change in absorbance due to the oxidation of cytochrome c was monitored at 550 nm. A parallel reaction in the presence of 300 µM KCN was also performed. The specific activity of complex IV was calculated by subtracting the KCN-resistant activity (with KCN) from the total complex IV activity (without KCN). The activities of complex III and ATP synthase were determined using the mitochondrial complex III activity assay kit (Biovision, #K520) and the ATP synthase enzyme activity microplate assay kit (Abcam, #109714), respectively, according to manufacturer's instructions.

**Intracellular ROS production and oxidative stress detection.** ROS generation was measured using the probe 2′,7′-dichlorodihydrofluorescein diacetate (H2DCFDA). Upon cleavage of the acetate groups by intracellular esterases and oxidation, the non-fluorescent H2DCFDA is converted to the highly fluorescent 2′,7′-dichlorofluorescein. Mitochondrial superoxide levels were measured using MitoSOX red (Thermo Fisher Scientific, #M36008). $10^6$ BTSCs were seeded in a $25 \, \text{cm}^2$ flask. Following 24 h of seeding, BTSCs were chemically dissociated using Accumax and loaded with 2 µM H2DCFDA or 1 µM MitoSOX red diluted in BTSC media. After 30 min of incubation (37 °C, 5% $CO_2$), the cells were harvested, washed, and resuspended in PBS containing 0.2% BSA. Fluorescence was assessed by flow cytometry (BD FACS CantoII). Data were analyzed using the FlowJo software. To measure superoxide levels in the tumor tissues, freshly isolated, frozen, and non-fixed tumor xenografts were sectioned by a cryostat at 8 µm. Sections were incubated with 10 µM MitoSOX red for 30 min at 37 °C, after which the MitoSOX solution was removed, and slides were imaged using a 40× objective on an Olympus IX83 microscope[70]. Fluorescence intensity was quantified using ImageJ software. For OxyIHC staining, freshly isolated tumor xenografts were sectioned by cryostat at 8 µm. Sections were fixed in Methacarn fixative solution (10% glacial acetic acid, 30% trichloromethane, 60% methanol) at 4 °C overnight. Staining was performed according to OxyIHC oxidative stress detection kit protocol (Millipore, #S7450).

**Seahorse bioenergetic analysis.** To measure OCR as an indication of metabolic activity, a Seahorse XFe96 flux Analyzer (Agilent Technologies) was used. BTSCs were chemically dissociated using Accumax prior to plating. XFe assay media (Agilent Technologies, 102353-100) supplemented with 2 mM sodium pyruvate (Sigma Aldrich, #P2256), 2 mM glutamine (Sigma Aldrich, #G3126), and 25 mM glucose was used. Cells were plated at a concentration of $0.9 \times 10^5$ cells/well in poly-D-lysine coated XFe 96-well plates (Agilent Technologies, #102353-100). The plates were then spun down and incubated at 37 °C in a 0% $CO_2$ incubator for 45 min. Following incubation, the mitochondrial stress test was performed using the following concentrations of toxins; 1.0 µM oligomycin, 2.0 µM carbonyl cya-nide-4-(trifluoromethoxy) phenylhydrazone (FCCP), 1.0 µM rotenone, and anti-mycin A (Agilent Technologies, #103015-100). The Seahorse flux analyzer wave software (V 2.3.0) was set for 3-min mixing time and 3-min measuring time. Toxins were added sequentially following every third measurement.

For ligand treatment assays, cells were pre-plated on 6-well plates and treated with 10 ng/mL OSM (Cell Signalling, #5367). Following treatment, cells were chemically dissociated and plated in the XFe assay media, as described above.

CGNs were plated on poly-D-lysine coated XFe 96-well plates at a concentration of $0.2 \times 10^6$ cells/well. The mito stress test assay was run 24 h after plating. Cells were treated with 1 µg/mL OSM (Cell Signalling, #5371) at the time of plating and media was changed to XFe assay media containing 1 mM sodium pyruvate, 2 mM glutamine, and 25 mM glucose 1 h prior to running the assay. 15 min after changing the media, the plate was transferred to a 0% $CO_2$ incubator at 37 °C for 45 min. 4.0 µM oligomycin, 2.0 µM FCCP, 1.0 µM rotenone, and antimycin A were used, as described above. Data were analyzed using Seahorse Wave Desktop 2.6 Software (Agilent).

**Immunofluorescence.** For immunostaining, BTSCs were plated on Lab-Tek II, CC2-treated chamber slide system (Thermo Fisher Scientific, #154941) in media containing 10% FBS, for 30 min. Cells were washed with PBS and fixed with 4% paraformaldehyde for 15 min at room temperature. Next, cells were permeabilized with 0.2% Triton-X (Sigma Aldrich, #T8787) for 10 min and blocked for 1 h with 5% normal donkey serum (NDS) in 1X-PBS. The cells were then incubated overnight at 4 °C with primary antibodies to ATPIF1 (1:100, Invitrogen, #A-21355) and OSMR (1:50, Abnova, #H00009180-D01P) diluted in 5% NDS-1X PBS. Cells were washed with PBS and then incubated with secondary Alexa fluor 488 goat anti-rabbit (1:500, Cell Signalling, #4412 s) and 594 goat anti-mouse (1:500, Cell Signalling, #8890) antibodies for 1 h. 2 µg/mL DAPI (Thermo Fisher Scientific, #D1306) was used to detect nuclei and ProLong Gold Antifade Mountant (Thermo Fisher Scientific, #P36934) was used for mounting. Images were captured using a 63X objective on a laser scanning confocal microscope (ZEISS LSM 800).

**Duolink proximity ligation assay.** PLAs were performed using a Duolink In Situ Red Starter Kit (Sigma, #DUO92101) according to the manufacturer's instructions. Briefly, BTSC were plated on Lab-Tek II, CC2-treated chamber slides in media containing 10% FBS, for 30 min. Cells were washed with PBS and fixed with 4% paraformaldehyde for 15 min at room temperature. For double labeling of the PLA signal and the MitoTracker, BTSCs were incubated for 30 min at 37 °C with 0.5 µM MitoTracker Deep Red FM (Thermo Fisher Scientific, #M22426) before fixation. Next, cells were permeabilized with 0.2% Triton-X (Sigma Aldrich, #T8787) for 10 min, blocked using Duolink blocking solution, and then incubated with primary antibodies at 4 °C overnight. After washing, the oligonucleotide (Minus and Plus)-conjugated secondary antibodies were added and incubated for 1 h at 37 °C. Subsequently, cells were washed and incubated with the ligation solution for 30 min at 37 °C. The ligated nucleotide circles were amplified using polymerase via the addition of the amplification solution followed by incubation for 100 min at 37 °C. The slides were washed briefly, and Duolink In Situ Mounting Medium with DAPI (DUO82040, Sigma) was added to each sample to stain the nuclei. The visualized fluorescence PLA signals were captured using a 63X objective on a laser scanning confocal microscope (ZEISS LSM 800).

**Fluorescence recovery after photobleaching assay.** GFP-OSMR expressing BTSC73 were cultured on coverslip II cell culture chamber slides (Sarstedt, #946190802). Cells were then incubated for 30 min at 37 °C with 0.2 µM Mito-Tracker Deep Red FM for mitochondrial staining. FRAP was performed using a laser scanning confocal microscope (ZEISS LSM 800). Briefly, mitochondrial-targeted GFP-OSMR was photobleached to ~98% of the initial signal using a brief pulse of high-intensity 488 nm laser illumination with 10 iterations. Regions of interest (ROIs) were imaged, using a 63X objective, before and after photo-bleaching at the indicated time points.

**Gene expression analysis.** Total RNAs were isolated from cells using TRIzol reagent (Invitrogen) according to the manufacturer's instructions. RNAs were then subjected to reverse transcription using the SuperScript III First-Strand cDNA synthesis system (Thermo Fisher Scientific, #11904018). Real-time quantitative polymerase chain reaction (RT-qPCR) was performed using the fluorescent dye SYBR Green (Life Technologies, #A25742). mRNA expression levels were then normalized to the housekeeping gene beta-glucuronidase (GUSB).

The following qPCR primers were used:
*SIRT3*-Fwd: CCAGAGGTTCTTGCTGCATG
*SIRT3*-Rev: CTCGGTCAAGCTGGCAAAAG
*ATP5B*-Fwd: CTGTACAGGCTATCTATGTGCC
*ATP5B*-Rev: GAGAGGTGGAGTCTAGAGGATC
*VDAC3*-Fwd: ATGGACTTACCTTCACCCAGAA
*VDAC3*-Rev: TCAAGAGTCAGTTTCAACCCTTC
*ATP6VOC*-Fwd: CCGGAGCAGATCATGAAGTC
*ATP6VOC*-Rev: TGTCGTCATTCAGGGAGTTG
*GUSB*-Fwd: GCGTTCCTTTTGCGAGGAGA
*GUSB*-Rev: GGTGGTATCAGTCTTGCTCAA
*OSMR*-Fwd: ACTGGAACCTGCCACAGAGT
*OSMR*-Rev: TCCAAGCTCACAATTCTCCA.

**Extracellular lactate assay.** Extracellular lactate was measured using an Amplite Colorimetric L-Lactate Assay Kit (AAT Bioquest, #13815). Briefly, supernatants from confluent flasks were diluted at 1:100, and 50 µL of the diluted media was added to 50 µL NAD containing assay buffer in a 96-well plate. The plate was

incubated at room temperature for 2 h and absorbance was measured using a CLARIOstar high-performance monochromator multimode microplate reader at 575 nm/605 nm. L-lactate standard curve is used to determine lactate concentrations in the samples.

**Stereotaxic injections and bioluminescent imaging**. All animal experiments were conducted under the institutional guidelines and were approved by McGill University Animal Care Committee (UACC). Housing room temperature and relative humidity were adjusted to 22.0 ± 2.0 °C and 55.0 ± 10.0%, respectively. The light/dark cycle was adjusted to 12 h lights-on and 12 h lights-off. Autoclaved water and irradiated food pellets (Tecklad 2918) were given ad libitum. For intracranial injections, $3 \times 10^5$ luciferase-expressing *OSMR* CRISPR and control BTSCs were stereotactically implanted into the right striata (0.8 mm lateral to the bregma, 1 mm dorsal and 2.5 mm from the pial surface) of randomized 7-week-old male SCID mice. Five and ten days following injections, mice received 4 Gy of IR using the X-Ray Irradiation System (Faxitron MultiRad 225). To examine tumor volume, the animals were intraperitoneally injected with 200 μL of 15 mg/mL D-luciferin (Thermo Fisher Scientific, #88292), anesthetized with isoflurane inhalation, and subjected to bioluminescence imaging with a CCD camera (IVIS, Xenogen) on a weekly basis. All bioluminescent data were collected and analyzed using IVIS software.

**Statistical analysis**. Statistical analysis was performed using ANOVA and Student's *t*-test, with the aid of GraphPad software 7. Two-tailed and unpaired t-tests were used to compare two conditions. One-way ANOVA with Tukey's or Dunnett's post hoc analyses were used for analyzing multiple groups. Data are shown as mean with standard error of mean (mean ± SEM). The log-rank test was used for statistical analysis in the Kaplan-Meier survival plot. *p*-Values of equal or less than 0.05 were considered significant and were marked with an asterisk on the histograms. *p*-Values of less than 0.05 are denoted by *, *p*-values of less than 0.01 are denoted by **, and *p*-values of less than 0.001 are denoted by *** on the histograms.

**Reporting Summary**. Further information on research design is available in the Nature Research Reporting Summary linked to this article.

## Data availability

The data supporting the findings of this study are included in the article and supplementary files. Additional raw data is available from the corresponding author upon request.

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

## Acknowledgements

This work was supported by grants from the Canadian Institute of Health Research # PJT 148986, PJT 145449, PJT 162198, and The Brain Tumor Charity, #497225 to AJA. AJA is an Fonds de la recherche en santé du Québec (FRQS) scholar in Glioblastoma Biology. AB is supported by an FRQS postdoctoral fellowship. We thank Dr. Samual Weiss at the University of Calgary for sharing BTSC73, 147, 12, and Dr. Keith Ligon at Harvard Medical School for the generation of BTSC112, 145, and 172. We thank Christian Young at the Lady Davis Institute for Medical Research – Jewish General Hospital – core facility for help with the Fluorescence-Activated Cell Sorting (FACS). We thank staff at the Lady Davis Institute Animal Core Facility for assistance with studies involving mice. We thank Perrine Gaub for technical assistance with the Seahorse bioenergetic analysis, and Felicia Lazure for technical help in IR experiments.

## Author contributions

Performed experiments: A.S., A.B., M.L. and A.J.-A.; Designed experiments and analyzed data: A.S., A.B., J.-S.J., V.D.S. and A.J.-A.; Wrote the paper: A.S., A.B. and A.J.-A.; Operation of seahorse facility: E.H.; Conceived the research program and provided funding and mentorship: A.J.-A.

## Competing interests

The authors declare no competing interests.
