## [Peer Review File · Nature Communications]

Reviewers' Comments:

Reviewer #1:

Remarks to the Author:

Sharanek and colleagues describe very convincingly that the cytokine receptor for Oncostatin M (OSMR) is localized in mitochondria of brain tumor stem cells (BTSCs) and regulates oxidative phosphorylation and thereby confers resistance to ionizing radiation. This was done by the help of western blot of subcellular fractions and by performing knockdown experiments. They identified interaction partners by IP-MS and were able to show that OSM regulates cellular respiration independent of EGFRvIII and that a knockdown of OSMR sensitizes BTSCs to irradiation in vitro and in vivo lowering sphere and tumor growth. Data are novel and impressive. However, in extended data fig 7 the authors describe a similar effect on primary CGN cells lowering the translational relevance of targeting OSMR. A few other criticisms are listed below:

- 1.) Extended data fig. 6: Did authors also analyze other brain cells? Glial cells would have been of particular interest. How was the purity of CGNs confirmed? What was the purity?
- 2.) Fig. e-f: Why are control values in f) higher as compared to e)?
- 3.) Fig. 3: Not only sphere numbers but also sphere sizes should be assessed. Pictures in a) and d) should show a larger area and more representative spheres.
- 4.) Page 8, last sentence: This is an overinterpretation. It has not been shown that especially slow-cycling BTSCs have been eliminated.
- 5.) For some of the western blot pictures it appears as if the samples were not run on the same gel because especially for OSMR there is often a lot of background. Please comment.
- 6.) Fig. 3f and g: these graphs should contain data with and without IR like in 3b, c and e).
- 7.) Glioblastoma are not any more called glioblastoma multiforme but simply glioblastoma. Please correct.
- 8.) Extended data, fig. 5a-b: Did the authors also test other time points and tissues?

Reviewer #2:

Remarks to the Author:

The contribution by Sharanek and colleagues adds on a previous contribution (Nat. Neuroscience 2016) that showed that the suppression of the receptor for oncostatin M (OSMR) prevents proliferation and tumor growth of glioblastoma cells and human xenografts in mice. OSMR is a co-receptor of the plasma membrane EGFRvIII receptor. In this manuscript they put forward the puzzling idea that OSMR is localized in mitochondria of brain tumor stem cells (BTSC) and in mouse liver and neurons and is responsible for the activation of mitochondrial respiration by interaction with subunits of complex I of the respiratory chain. In addition, they show that knockdown of OSMR increases the response of brain xenografts to ionizing radiation (IR). My major concerns with this piece of work are the methodologies used to claim the mitochondrial localization of OSMR and details of the mechanism of action of OSMR in mitochondrial respiration as detailed in the following comments. Addressing these concerns is required before the manuscript could be considered for publication in Nat. Commun.

Major points.

1. Fig. 1a-f and Extended Fig. 5: It is claimed that OSMR is present in mitochondria of BTSCs, liver and brain. However, membrane fractions are known to co-purify with mitochondria. These blots should include biomarkers of plasma and ER membranes to exclude the contamination of plasma/ER membranes in the Mito-fraction. Plasma membrane as such needs to be included as a fraction of the separation and a blot of EGFRvIII could illuminate the findings.
2. Fig. 1 g-h: The proteinase K/Triton X100 experiment supports that OSMR is protected in a membranous compartment but not necessarily mitochondria. They should address a specific import assay of OSMR in isolated mitochondria.
3. Fig. 1 k-n, Fig.1 r-s and Supp. Table 1: Blots of IPs and the "interactome" study by IP-LC-MS/MS are carried out from solubilized proteins from whole cellular lysates where the abundant mitochondrial proteins are highly represented. The IPs should be performed from solubilized

proteins from isolated mitochondria. Moreover, the interactome data is really misleading they only show a set of mitochondrial proteins out of the more than thousand proteins that are obtained by this basic approach. This sort of studies need to be quantitatively carried out using SILAC. For instance, in their case, by labeling with heavy and light amino acids the OSRM and knockdown-OSRM cells, respectively. For details in the way to procedure see the example in He et al., PNAS (2018) 115, 2988-2993.

4. The mechanisms to explain the upregulation of mitochondrial respiration mediated by OSMR either alone or in response to the incubation of the cells with OSM (Figs. 2a,e and Extended Figs. 2 and 3) are not explained and far from clear. The major downstream signaling pathways that are activated in response to OSM are JAK/STAT, Ras/Raf/MAPK and PI3K pathways which could be modulating the activity and phosphorylation status of oxidative phosphorylation complexes, these facts are not taken into consideration and, in contrast, they emphasize the potential interaction of OSMR with NDUFS1, a matrix protruding subunit of complex I, which is unconvincing (see point 3) and by itself does not explain the increase in respiration. They should determine the enzymatic activities of respiratory chain complexes and eventually the differential phosphorylation of immunocaptured respiratory complexes.

Minor points:

1.- The differential redox stress of the xenografts in response to IR should be documented.

We thank the reviewers for their insightful comments and suggestions. In the revised manuscript, we have addressed each of the points raised by the reviewers by performing new experiments, followed by revision to the text as described in the point by point response, below. We believe that the revised manuscript has significantly improved.

Reviewer #1 (Remarks to the Author):

Sharanek and colleagues describe very convincingly that the cytokine receptor for Oncostatin M (OSMR) is localized in mitochondria of brain tumor stem cells (BTSCs) and regulates oxidative phosphorylation and thereby confers resistance to ionizing radiation. This was done by the help of western blot of subcellular fractions and by performing knockdown experiments. They identified interaction partners by IP-MS and were able to show that OSM regulates cellular respiration independent of EGFRvIII and that a knockdown of OSMR sensitizes BTSCs to irradiation in vitro and in vivo lowering sphere and tumor growth. Data are novel and impressive. However, in extended data fig 7 the authors describe a similar effect on primary CGN cells lowering the translational relevance of targeting OSMR. A few other criticisms are listed below:

We thank the reviewer for stating that our study is “novel”, “impressive” and we “describe very convincingly the cytokine receptor for Oncostatin M (OSMR) is localized in mitochondria of brain tumor stem cells (BTSCs) and regulates oxidative phosphorylation and thereby confers resistance to ionizing radiation”.

With respect to the CGN data, we would like to clarify that due to low OSMR expression levels, all the experiments in the CGN cultures (including Supplementary Fig. 7a), were carried out upon forced expression of OSMR and addition of the ligand OSM in order to uncouple the role of OSMR in cell proliferation versus mitochondrial respiration. By using this model, we provide experimental evidence that OSMR regulation of OCR and its role in conferring resistance to DNA damage-induced cell death is independent of cell proliferation or other oncogenic signalling pathways that may converge on the mitochondria. This detail has been added to page 11 as follow:

“Our data confirm that OSM/OSMR regulation of mitochondrial respiration is independent of its role in cell proliferation, and this mechanism is operational in post-mitotic neurons upon forced induction of OSM/OSMR signalling pathways.”

We would like to clarify that OSMR is an attractive therapeutic target for glioblastoma for multiple reasons:

First, OSMR is highly upregulated in glioblastoma compared to low grade glioma or normal tissue, and increased expression of OSMR is an important predictor of glioblastoma patient survival (Jahani-Asl et al, Nature Neuroscience 2016, Natesh et al, Neoplasia 2015, Xu et al 2018, BMC Med Genomics, Cao et al, Cancer Biol Med 2019).

Second, we and others have demonstrated that OSMR knockout mice are viable, healthy and fertile (Hams et al, J Immunol 2008, Tanaka et al, Blood 2003, Tonkin et al, Osteoarthritis and Cartilage 2014, Zhang et al, J Lipid Research 2017, Burban and Jahani-Asl et al, unpublished). Therefore, targeting of OSMR is not expected to result in unfavourable side effects.

Third, OSMR is highly upregulated in brain tumour stem cells (Jahani-Asl et al, Nature Neuroscience 2016). In the present manuscript, we report that OSMR regulates cellular respiration and confers BTSC resistance to ionizing radiation. Thus, OSMR is an attractive therapeutic target to deplete cancer stem cells in the tumour mass and facilitate the response of glioblastoma tumours to IR therapy.

1.) Extended data fig. 6: Did authors also analyze other brain cells? Glial cells would have been of particular interest. How was the purity of CGNs confirmed? What was the purity?

We have analyzed other brain cells. As suggested by the reviewer, we provide analysis of glial cells, in which we demonstrate that similar to BTSCs (Fig. 2) and post-mitotic primary neurons [Extended data Fig. 6 (i.e. Supplementary Fig. 5d in the revised manuscript)], OSMR interacts with mtHSP70, TIM44, NDUFS1 and NDUFS2 in EGFRvIII-expressing astrocytes (**New Supplementary Fig. 1c**). CGN cultures were performed using our established protocol (Laaper et al, JoVE 2018, Jahani-Asl et al, JBC 2007, Jahani-Asl et al, JBC 2011, Jahani-Asl et al, Human Molecular Genetics 2015). These cultures represent a homogenous pure (> 90%) neuronal population.

2.) Fig. e-f: Why are control values in f) higher as compared to e)?

For all the LDA plots, the experiments were carried out on different passage numbers of BTSCs for different experiments and multiple replicates were performed to take into account experimental variations. In the revised manuscript, we have repeated the LDA plots with and without IR in OSMR KD or control BTSCs by performing the experiments in parallel on similar BTSC passage numbers, and we obtained comparable values (**New Fig. 5f** and **New Supplementary Fig. 6c**). Importantly, in the revised manuscript, all the LDA plots have been validated using ELDA analysis on similar BTSCs passage numbers in parallel, as described in our response to point #6.

3.) Fig. 3: Not only sphere numbers but also sphere sizes should be assessed. Pictures in a) and d) should show a larger area and more representative spheres.

As requested by the reviewer, we have also assessed the sphere size for BTSC73 and BTSC147 in response to IR as follow:

- ◆ Impact of OSMR CRISPR on sphere size in irradiated and control BTSC73 in which we find that OSMR CRISPR in combination with IR significantly reduced sphere size (**New Supplementary Fig. 6e**).
- ◆ Impact of siOSMR on sphere size in irradiated and control BTSC73 in which we demonstrate that transient KD of OSMR in combination with IR significantly reduced sphere size (**New Supplementary Fig. 6f**).
- ◆ Impact of siOSMR on sphere size in irradiated and control BTSC147 in which we show that the transient knockdown of OSMR in combination with IR significantly reduced sphere size (**New Supplementary Fig. 6g**).

As requested by the reviewer, in the revised manuscript we have provided a larger representative area corresponding to the counts (**New Supplementary Fig. 6a** and **New Supplementary Fig. 6b**).

4.) Page 8, last sentence: This is an overinterpretation. It has not been shown that especially slow-cycling BTSCs have been eliminated.

As requested by the reviewer, we have revised the paragraph in the revised version of the Discussion on page 16 as follow:

“In conclusion, our study has uncovered an important mechanism whereby cytokine signalling alters BTSC metabolism and confers glioblastoma resistance to therapy. OSM/OSMR targeted therapies are promising in eliminating IR-resistant BTSCs in the tumour mass, impairing mitochondrial function, and improving response to IR.”

5.) For some of the western blot pictures it appears as if the samples were not run on the same gel because especially for OSMR there is often a lot of background. Please comment.

OSMR expression levels are low in wild type tissues used in this study, thus long exposure of the blots results in a lot of background.

All samples including input and IP were run on the same gel. Importantly, we have presented the same exposure for both input and IP lanes, rather than cropping different exposures for each. Here, we present an example of the original blot. Blue rectangles represent the areas that were cropped. All exposures are taken using ChemiDoc™ Touch Imaging System (1708370) by Bio-Rad.

6.) Fig. 3f and g: these graphs should contain data with and without IR like in 3b, c and e).

As requested by the reviewer, we present data with and without IR in **New revised Fig. 5f** and **New Supplementary Fig. 6c** of the revised manuscript. In addition to the LDA assays, we have performed new experiments with and without IR using ELDA as an additional readout and the data is presented in the revised manuscript as described below:

- ◆ Impact of OSM ligand on irradiated and control BTSC73, in which we show that OSM confers BTSC73 resistance to IR (**New Fig. 5c**).
- ◆ Impact of OSM ligand on irradiated and control BTSC147, in which we show that OSM confers BTSC147 resistance to IR (**New Fig. 5d**).
- ◆ Impact of *OSMR* CRISPR in response to IR in BTSC73, in which we show that CRISPR of *OSMR* facilitates the response of BTSC73 to IR (**New Fig. 5g**).
- ◆ Impact of *OSMR* siRNA in BTSC147, in which we show that transient KD of *OSMR* facilitates the response of BTSC147 to IR (**New Fig. 5h**).
- ◆ Impact of *OSMR* siRNA in BTSC73, in which we show that KD of *OSMR* facilitates the response of BTSC73 to IR (**New Supplementary Fig. 6d**).

Data with and without IR are presented.

In summary, using both ELDA and LDA, in different BTSCs, we find that activation of OSM/*OSMR* signalling pathway confers BTSC resistance to IR.

7.) Glioblastoma are not any more called glioblastoma multiforme but simply glioblastoma. Please correct.

As requested by the reviewer, we have replaced all the abbreviations (GBM) in the text to glioblastoma.

8.) Extended data, fig. 5a-b: Did the authors also test other time points and tissues?

We have tested *OSMR* protein expression levels in the brain tissue at different time point. Representative *OSMR* blots are presented in **New Supplementary Fig. 5a**. *OSMR* expression levels significantly declines post 2 months. We also tested *OSMR* levels in 5 different organs i.e. brain, liver, spleen, kidney and heart and found that *OSMR* levels were barely detectable in spleen, kidney and heart (data not shown). Thus, we performed the fractionation in the brain and liver from 2 months old mice.

Reviewer #2 (Remarks to the Author):

The contribution by Sharanek and colleagues adds on a previous contribution (Nat. Neuroscience 2016) that showed that the suppression of the receptor for oncostatin M (*OSMR*) prevents proliferation and tumor growth of glioblastoma cells and human xenografts in mice. *OSMR* is a co-receptor of the plasma membrane EGFRvIII receptor. In this manuscript they put forward the puzzling idea that *OSMR* is localized in mitochondria of brain tumor stem cells (BTSC) and in mouse liver and neurons and is responsible for the activation of mitochondrial respiration by interaction with subunits of complex I of the respiratory chain. In addition, they show that knockdown of *OSMR* increases the response of brain xenografts to ionizing radiation (IR). My major concerns with this piece of work are the methodologies used to claim the mitochondrial localization of *OSMR* and details of the mechanism of action of *OSMR* in mitochondrial respiration as detailed in the following comments. Addressing these

concerns is required before the manuscript could be considered for publication in Nat. Commun.

We thank Prof. Cuezva for careful consideration of our manuscript and many excellent suggestions that have profoundly improved our manuscript. In the revised manuscript, we report a novel function

for OSMR beyond the regulation of EGFRvIII/STAT3 signalling pathway and independently of its role in cell proliferation. We provide compelling evidence that OSMR is translocated to the mitochondria to promote mitochondrial respiration and confer resistance of glioblastoma to IR therapy. As requested, we have added new methodologies and experiments which together have greatly strengthened the conclusion.

Major points.

1. Fig. 1a-f and Extended Fig. 5: It is claimed that OSMR is present in mitochondria of BTSCs, liver and brain. However, membrane fractions are known to co-purify with mitochondria. These blots should include biomarkers of plasma and ER membranes to exclude the contamination of plasma/ER membranes in the Mito-fraction. Plasma membrane as such needs to be included as a fraction of the separation and a blot of EGFRvIII could illuminate the findings.

As requested by Prof Cuezva, all the immunoblots of fractionated BTSCs, liver and brain were probed for markers of plasma membrane (sodium potassium ATPase) and ER membrane (calnexin). We show that the mitochondrial fractions were of very high purity with no cross contamination with ER or plasma membrane fractions (**New revised Fig. 1 a-d**, and **New revised Supplementary Fig. 5 b-c**). Since EGFR and EGFRvIII are both reported to localize to the mitochondria (Che et al., Oncotarget 2015, Wang et al., Oncogenesis 2017, Cao et al., Mol Cancer 2011, Boerner et al., Mol Cell Biol 2004), they are not serving as an appropriate control. Therefore, we probed the blots for sodium potassium ATPase and calnexin. Furthermore, confocal imaging analysis were provided to demonstrate that OSMR was found in puncti with the mitochondrial markers (**Revised Figure 1e-f**).

In the revised manuscript, we have performed **two new assays**: **1)** proximity ligation assays (PLA) in multiple patient-derived BTSCs to demonstrate *in situ* interaction of OSMR with the mitochondria (**New Fig. 1g-i**) as well as different components of PAM complex (**New Fig. 2g-i**) and complex I of ETC (**New Fig. 2p-s**); **2)** We performed fluorescence recovery after photobleaching (FRAP) analysis in which we show that OSMR is recruited to the mitochondria following photobleaching (**New Fig. 1j**). Please also see our response to point # 2.

Together, using cell fractionation, high resolution confocal imaging, PLA analysis and double-labelling of PLA signal with the MitoTracker, and FRAP assay, we establish that OSMR is present in the mitochondria.

2. Fig. 1 g-h: The proteinase K/Triton X100 experiment supports that OSMR is protected in a membranous compartment but not necessarily mitochondria. They should address a specific import assay of OSMR in isolated mitochondria.

First, as requested by Prof. Cuezva, we designed a FRAP assay in which the BTSCs were transduced with GFP-OSMR fusion protein, co-labelled with the MitoTracker. BTSC were subjected to GFP photobleaching, followed by time lapse imaging to assess the rate of recovery of the GFP signal in the mitochondria in real time. Using this assay, we show that OSMR is recruited to the mitochondria in human BTSCs (**New Fig. 1j**).

Second, we have performed PLA analysis as well as double labelling of PLA interaction signal with the Mitotracker as follow:

- ◆ OSMR/ATPIF1 in BTSC73 (**New Fig. 1g, New Fig. 1i**)
- ◆ OSMR/ATPIF1 in BTSC147 (**New Fig. 1h**)
- ◆ OSMR/mtHSP70 in BTSC73 (**New Fig. 2g, New Fig. 2i**)
- ◆ OSMR/mtHSP70 in BTSC147 (**New Fig. 2h**)
- ◆ OSMR/NDUFS1 and OSMR/NDUFS2 in BTSC73 (**New Fig. 2p, New Fig. 2r-s**)
- ◆ OSMR/NDUFS1 and OSMR/NDUFS2 in BTSC147 (**New Fig. 2q**)

Third, we repeated all the co-immunoprecipitation experiments on the mitochondrial fractions (**New Fig. 2d, New Fig. 2f, New Fig. 2m and New Fig. 2o**) (as described in response to point 3).

These new analyses together with our previous results establish the presence of OSMR in the mitochondria.

3. Fig. 1 k-n, Fig.1 r-s and Supp. Table 1: Blots of IPs and the “interactome” study by IP-LC-MS/MS are carried out from solubilized proteins from whole cellular lysates where the abundant mitochondrial proteins are highly represented. The IPs should be performed from solubilized proteins from isolated mitochondria. Moreover, the interactome data is really misleading they only show a set of mitochondrial proteins out of the more than thousand proteins that are obtained by this basic approach. This sort of studies need to be quantitatively carried out using SILAC. For instance, in their case, by labeling with heavy and light amino acids the OSRM and knockdown-OSRM cells, respectively. For details in the way to procedure see the example in He et al., PNAS (2018) 115, 2988-2993.

We agree with Prof. Cuezva that among many different screening approaches of immunoprecipitation (IP) including IP with tagged proteins, BioID, SILAC and so on, the proposed SILAC protocol is interesting. However, regardless of which screening approaches used, the data must be validated using specific interaction assays. In our manuscript, we have validated the IP-MS interactions using IP-WB of whole cell lysates (**Fig. 2c, Fig. 2e, Fig. 2l, Fig. 2n, Supplementary Fig. 5d**), new experiments in which we performed IP-WB on purified mitochondrial fractions (**New Fig. 2d, New Fig. 2f, New Fig. 2m, New Fig. 2o, New Supplementary Fig. 1c**), new PLA analysis in multiple cell lines (**New Fig. 1g-i, New Fig. 2g-i, New Fig. 2p-s**). Furthermore, in our IP we did not use sticky tags (e.g. FLAG tagged proteins which are commonly used). Rather, we took advantage of an OSMR antibody that we previously established to be working for endogenous IP (Jahani-Asl et al, Nature Neuroscience 2016). Thus, we were able to benefit from this system to pull down OSMR potential binding partners endogenously, without relying on tagged proteins that could be sticky and pull down a lot of false positives. Thus, we obtained far less than 1000 potential binding proteins.

In conclusion, using IP-Western blot on both whole cell lysates and purified mitochondrial fractions as well as PLA analysis, we have validated the reported interactions endogenously and *in situ* in multiple patient-derived cell lines.

4. The mechanisms to explain the upregulation of mitochondrial respiration mediated by OSMR either alone or in response to the incubation of the cells with OSM (Figs. 2a,e and Extended Figs. 2 and 3) are not explained and far from clear. The major downstream signaling pathways that are activated in response to OSM are JAK/STAT, Ras/Raf/MAPK and PI3K pathways which could be modulating the activity and phosphorylation status of oxidative phosphorylation complexes, these facts are not taken into consideration and, in contrast, they emphasize the potential interaction of OSMR with NDUFS1, a matrix protruding subunit of complex I, which is unconvincing (see point 3) and by itself does not explain the increase in respiration. They should determine the enzymatic activities of respiratory chain complexes and eventually the differential phosphorylation of immunocaptured respiratory complexes.

As requested by Prof Cuezva, we first addressed whether JAK/STAT, MAPK and PI3K/AKT pathways are activated in response to OSM in patient derived glioma stem cells.

We determined that:

- ◆ p42/44 MAPK is phosphorylated in response to OSM treatment, and its phosphorylation is inhibited by PD0325901 (**New Supplementary Fig. 4a**).
- ◆ AKT is phosphorylated by OSM treatment, and its phosphorylation is inhibited by LY294002 (**New Supplementary Fig. 4b**).
- ◆ STAT3 is phosphorylated by OSM treatment, and its phosphorylation is inhibited by WP1066 (**New Supplementary Fig. 4c**).

Second, we examined oxygen consumption rate (OCR) by performing Seahorse experiments as follows:

- ◆ BTSC73 cells were treated with OSM or OSM + p42/44 MAPK inhibitor (PD0325901) (**New Fig. 4e**).
- ◆ BTSC73 cells were treated with OSM or OSM + PI3K inhibitor (LY294002) (**New Fig. 4f**).
- ◆ BTSC73 cells were treated with OSM or OSM + JAK/STAT3 inhibitor (WP1066) (**New Fig. 4g**).

Addition of OSM induced a significant increase in respiration. While inhibition of p42/44 MAPK did not show any apparent effect on OSM-induced increase in OCR, inhibition of either PI3K or STAT3 resulted in a partial decrease in OSM-induced mitochondrial respiration. We found that OSM robustly promotes mitochondrial respiration even after considering the effect of MAPK, PI3K/AKT, or STAT3 activation. These results, furthermore, support our previous findings whereby OSM/OSMR signaling significantly increased mitochondrial respiration in post-mitotic wild type primary neurons (**Fig. 4h**). The new data included in Fig. 4 are discussed on **pages 10-11**.

Third, as requested by Dr. Cuezva we measured the enzymatic activity of respiratory chain complexes as follow:

- ◆ Complex I activity in *OSMR* CRISPR compared to BTSC73 control (**New Fig. 3a**)
- ◆ Complex II activity in *OSMR* CRISPR compared to BTSC73 control (**New Fig. 3b**)
- ◆ Complex III activity in *OSMR* CRISPR compared to BTSC73 control (**New Fig. 3c**)
- ◆ Complex IV activity in *OSMR* CRISPR compared to BTSC73 control (**New Fig. 3d**)
- ◆ ATP synthase activity in *OSMR* CRISPR compared to BTSC73 control (**New Fig. 3e**)

Our data demonstrate that knockdown of *OSMR* resulted in a significant reduction of complex I, complex II, complex III and complex IV activities. However, no significant change in ATP synthase activity in *OSMR* CRISPR compared to CTL was observed. These data added to the **New Fig. 3 a-e** are discussed in the revised manuscript on **page 8** of the results and **page 15** of the discussion.

Minor points:

1.- The differential redox stress of the xenografts in response to IR should be documented.

Signed: José M Cuezva

As requested by Prof Cuezva oxidative stress in BTSC patient-derived tumour xenografts from CTL, *OSMR* CRISPR, IR and IR + *OSMR* CRISPR was assessed using MitoSOX staining. Our data show that ROS generation was significantly higher in IR + *OSMR* CRISPR tumour tissues compared to IR or *OSMR* CRISPR alone. These data are now added in **New Fig. 7f and New Supplementary Fig. 7b**.

Sincerely,
Dr. Arezu Jahani-Asl
McGill University

Reviewers' Comments:

Reviewer #1:

Remarks to the Author:

My concerns have been sufficiently expressed.

signed: Christel Herold-Mende

Reviewer #2:

Remarks to the Author:

The revised paper has been much improved showing the localization of OSMR in mitochondria of different cellular types. However, this reviewer does not find mechanistic explanation, or sufficiently convincing arguments, to explain the OSMR-mediated activation of mitochondrial respiration. Likewise, there is no mechanistic explanation to address the inhibition of the activities of the four respiratory complexes studied when OSMR is silenced.

Reviewer #1 (Remarks to the Author):

My concerns have been sufficiently expressed.

signed: Christel Herold-Mende

Reviewer #2 (Remarks to the Author):

The revised paper has been much improved showing the localization of OSMR in mitochondria of different cellular types. However, this reviewer does not find mechanistic explanation, or sufficiently convincing arguments, to explain the OSMR-mediated activation of mitochondrial respiration. Likewise, there is no mechanistic explanation to address the inhibition of the activities of the four respiratory complexes studied when OSMR is silenced.

Jose M Cuezva

We would like to thank Professor Christel Herold-Mende and Professor Jose M Cuezva for careful consideration of our revised manuscript, and that they state that the concerns have been sufficiently expressed/the manuscript is much improved. In response to Prof. Cuezva's request on explanations/arguments on OSMR-mediated mitochondrial respiration and regulation of the respiratory complex, we provide extensive data to show that OSMR interacts with the components of complex I, NDUFS1 and NDUFS2, and that deletion of OSMR phenocopies NDUFS1 knockdown (KD) effects. KD of NDUFS1 has been previously shown to impair the association of complex I into supercomplexes leading to impaired oxygen consumption rate and increased ROS (Lopez-Fabuel et al 2016, PNAS). We show that OSMR interacts with NDUFS1 and KD of *OSMR* leads to impaired oxygen consumption rate, increased ROS, and impairment of the activity of respiratory complexes. Our data supports a model whereby OSMR/NDUFS1 complex contributes to ROS buffering and increase in oxygen consumption rate. Consequently, NDUFS1/OSMR complex may facilitate the assembly and stability of supercomplexes. This is supported by our finding that although OSMR interacts with complex I, KD of *OSMR* impairs not only the enzymatic activity of complex I but significantly impairs complex II, III, and IV activities. An extended mechanistic explanation is provided in the Discussion section as follow:

"Complex I is the major source of mitochondrial ROS^{39,69-71}. We established that OSMR directly interacts with NDUFS1/NDUFS2 of complex I in the mitochondria of patient derived glioma stem cells. Loss of *NDUFS1* function is shown to impair OCR and lead to the accumulation of ROS^{39,72,73}. Furthermore, *NDUFS1/2* mutations are associated with complex I deficiency that is presented in different OXPHOS disorders⁷²⁻⁷⁵. Similar to *NDUFS1*, *OSMR* deletion impairs OCR and leads to excess ROS suggesting that loss of *OSMR* phenocopies *NDUFS1* KD effects. Our observations suggest a model whereby OSMR/NDUFS1 complex function in the same pathway to regulate mitochondrial respiration and buffer excess ROS.

The function of the mitochondrial respiratory chain involves the organization of the enzyme complexes into respirasomes or supercomplexes that are structurally interdependent^{70,76-79}. Therefore, defects in a single ETC component often produce combined enzyme deficiencies in patients⁸⁰⁻⁸⁴. The formation of supercomplexes has

been reported to be severely reduced in mutated *NDUFS1* patient⁸⁵. Furthermore, *NDUFS1* KD in primary cortical neurons was shown to decrease the assembly of complex I into supercomplexes³⁹. We found that although OSMR interacts with complex I, loss of OSMR impairs the enzymatic activities of all four complexes in the ETC. Our data support a model whereby OSMR may contribute to the assembly of supercomplex via a cross talk with *NDUFS1*.”